

# Understanding aerosol-cloud interactions through modelling the development of orographic cumulus congestus during IPHEx

Yajuan Duan[1], Markus D. Petters[2], and Ana P. Barros[1\$]

[1]Department of Civil and Environmental Engineering, Duke University, Durham, NC, USA

[2]Department of Marine Earth and Atmospheric Sciences, North Carolina State University, Raleigh, NC, USA

[\$]*Correspondence to*: Ana P. Barros (barros@duke.edu)

**Abstract.** A new cloud parcel model (CPM) including activation, condensation, collision-coalescence, and lateral entrainment processes is presented here to investigate aerosol-cloud interactions (ACI) in cumulus development prior to rainfall onset. The CPM was applied with surface aerosol measurements to predict the vertical structure of cloud development at early stages, and
the model results were compared with airborne observations of cloud microphysics and thermodynamic conditions collected during the Integrated Precipitation and Hydrology Experiment (IPHEx) in the inner region of the Southern Appalachian Mountains (SAM). Sensitivity analysis was conducted to examine the model response to variations in key ACI physical parameters. The condensation coefficient $a_c$ plays a governing role in determining the cloud droplet number concentration (CDNC), liquid water content (LWC), and droplet size distribution. Lower values of $a_c$ lead to higher cloud droplet number
concentrations, broader droplet spectra, and higher maximum supersaturation near cloud base. The simulated vertical structure of CDNC exhibits strong nonlinear sensitivity to entrainment strength and condensation efficiency illustrative of competitive interference between turbulent dispersion and activation processes. Further, simulated CDNC exhibits high sensitivity to variations in initial aerosol concentration at cloud base, but weak sensitivity to aerosol hygroscopicity. These findings provide new insights into determinant factors of convective cloud formation leading to mid-day warm season rainfall in complex
terrain.

## 1 Introduction

Atmospheric aerosols produced by dramatically increased industrialization and urbanization exert a large impact on the climate system and the hydrological cycle (Koren et al., 2008;Ramanathan et al., 2001;Tao et al., 2012). Aerosols influence the earth-atmosphere system primarily via two mechanisms: a radiative (direct) effect and a microphysical (indirect) effect
(Rosenfeld et al., 2008). The direct effect on the Earth's energy budget occurs via scattering and absorbing of shortwave and longwave radiation in the atmosphere, hence modulating the net radiation and climate (Haywood and Boucher, 2000;Ramanathan et al., 2001). The indirect effect is related to aerosols as cloud condensation nuclei (CCN) or ice nuclei (IN) that alter microphysical properties and consequently affect cloud radiative properties and precipitation efficiency (Jiang et al., 2008;Lohmann and Feichter, 2005;McFiggans et al., 2006). In particular, an increase in aerosol concentration results in



enhanced cloud droplet number concentration (CDNC), smaller average drop size, and increased cloud albedo (Twomey, 1977). Smaller cloud droplets are associated with lower collection and coalescence efficiency, slower drop growth and reduced precipitation, thus leading to longer cloud lifetimes (Albrecht, 1989;Andreae and Rosenfeld, 2008;Khain et al., 2005). Over complex terrain in California and Israel, Givati and Rosenfeld (2004) attributed a reduction in annual precipitation of 15–25%

to air-pollution aerosols from upwind urban areas. By comparing two scenarios of maritime and continental aerosols, Lynn et al. (2007) found that simulations with maritime aerosols with relatively lower aerosol number concentration yielded 30% more precipitation than continental aerosols over a mountain slope. Such local effects can translate into large spatial shifts in clouds and precipitation as aerosol-cloud interactions (ACI) inducing suppression of precipitation upwind could give rise to the enhancement of precipitation downwind (Muhlbauer and Lohmann, 2008), thus modifying the spatial distribution of

orographic precipitation, transferring precipitation from one watershed to another and thus strongly influencing the local and regional hydrology. Yang et al. (2016) examined the reasons for warm rain suppression due to increased air pollution in the Mt. Hua area in central China. They demonstrated that weakened valley-ridge circulations as a result of aerosol-radiation interactions and lower water vapor concentrations in the valley led to the suppression of convection and precipitation in the mountain. A study of thermally driven orographic clouds over a tropical island during the Dominica Experiment (DOMEX)

field campaign found that atmospheric moisture was the predominant constraint in cloud and precipitation formation over the aerosol effect, and the surface aerosol source has the strongest influence on precipitation under unfavourable environmental condition for cloud growth (Nugent et al., 2016).

Because of their multiscale nature and complex physics, the representation of physical and chemical processes related to clouds and precipitation in numerical models relies on parameterizations with varying degrees of uncertainties depending

on space-time model resolution (Khairoutdinov et al., 2005;Randall et al., 2003). For example, the characteristic time-scale of condensational growth of submicron-size droplets is on the order of 1 ms, and length scales of individual drops range from μm to cm (Pinsky and Khain, 2002), that is a scale gap of five to nine orders in magnitude with respect to the spatial resolution of cloud-resolving models (kms). Although detailed 2-D and 3-D models that explicitly resolve cloud formation and microphysical evolution to varying degrees of completeness have been reported in the literature (Fan et al., 2009;Leroy et al.,

2009;Muhlbauer et al., 2010), the wide range of length (μm–m) and time (ms–s) scales associated with aerosol-cloud-rainfall interactions poses significant challenges for model spatial and temporal resolution. For instance, analysis of high resolution (~ 1 km) numerical weather prediction (NWP) simulations in the SAM for various hydrometeorological regimes using different Weather Research and Forecasting (WRF) physical parameterizations concluded that the prediction of cloud development and cloud vertical microphysical structure are inadequate to capture the spatial and temporal resolution of precipitation rate and

precipitation microphysics at the ground (Wilson and Barros, 2015, 2017). In particular, six microphysical parameterization schemes available in WRF were examined to investigate the spatiotemporal evolution of low level moisture fields in the SAM under weak and strong synoptic conditions. However, the simulations could not capture persistent low-level clouds and fog (LLCF) and in particular the mid-day rainfall peak observed in this region (Duan and Barros, 2017b;Wilson and Barros, 2015). Furthermore, simulations exploring the use of different planetary boundary layer (PBL) parameterizations in WRF could not





replicate the observed vertical structure of LLCF, thus failing to reproduce the reverse orographic enhancement linked to seeder-feeder interactions (SFI), and consequently resulting in significantly lower rainfall intensities as compared to the surface observations (Duan and Barros, 2017b;Wilson and Barros, 2014, 2015, 2017).

An alternative modeling approach to investigate ACI at fine resolution is the cloud parcel model (CPM) to simulate aerosol activation and cloud droplet growth, as well as thermodynamic adaptation of ascending air parcels at μm and ms scales (Abdul-Razzak et al., 1998;Cooper et al., 1997;Flossmann et al., 1985;Jacobson and Turco, 1995;Kerkweg et al., 2003;Nenes et al., 2001;Pinsky and Khain, 2002;Snider et al., 2003). A synthesis of model formulation including spectral binning strategy, principal physical processes (i.e., condensational growth, collision-coalescence, entrainment), and key aspects of their numerical implementation is presented in Table 1 for CPMs frequently referred to in the peer-reviewed literature. In the past, process studies using CPMs targeted principally aerosol-CDNC closure between model simulations and field observations. For example, Conant et al. (2004) conducted an aerosol-cloud droplet number closure study against observations from NASA's Cirrus Regional Study of Tropical Anvils and Cirrus Layers–Florida Area Cirrus Experiment (CRYSTAL-FACE) using the adiabatic CPM by Nenes et al. (2001;2002) that solves activation and condensation processes only (see Table 1 for details). Using a condensation coefficient ($a_c$) value of 0.06, they reported that predicted CDNC was on average within 15% of the observed CDNC in adiabatic cloud regions. Fountoukis et al. (2007) used the same CPM as Conant et al. under extremely polluted conditions during the 2004 International Consortium for Atmospheric Research on Transport and Transformation (ICARTT) experiment. They found that the optimal closure of cloud droplet concentrations was achieved when the condensation coefficient was about 0.06. For marine stratocumulus clouds sampled during the second Aerosol Characterization Experiment (ACE-2), Snider et al. (2003) applied the UWyo parcel model (http://www.atmos.uwyo.edu/~jsnider/parcel/) to simulate condensation processes in adiabatic ascent (see Table 1) and experimented with various condensation coefficients in the range of 0.01–0.81. They hypothesized that the lower CDNC overestimation errors (20 to 30% for $a_c = 0.1$) in their CPM simulations could be mitigated by varying the condensation coefficient as a function of dry particle size instead of using one value for the entire distributions, but did not actually demonstrate this was the case. The condensation coefficient of water is a key ACI physical parameter in parcel models and has a strong influence on activation and droplet growth by condensation as it expresses the probability that vapour molecules impinge on the water droplet when they strike the air-water interface (McFiggans et al., 2006). Experimental measurements reviewed by Marek and Straub (2001) exhibit a strong inverse relationship between pressure and $a_c$ values ranging from 1000 hPa to 100 hPa and from 0.007 to 0.1, respectively (their Fig. 4). Chodes et al. (1974) measured condensation coefficients in the range of 0.02–0.05 with a mean of 0.033 from measurements of individual droplets grown in a thermal diffusion chamber for four different supersaturations. Ganier et al. (1987) repeated Chodes et al.'s experiments and found that the average condensation coefficient is closer to 0.02 after correcting their supersaturation calculations. Shaw and Lamb (1999) conducted extensive simultaneous measurements of the condensation coefficient and thermal accommodation coefficients ($a_T$) for individual drops in a levitation cell and reported values for $a_c$ and $a_T$ in the ranges of 0.04–0.1 and 0.1–1 with most probable values of 0.06 and 0.7, respectively. Using adiabatic CPMs



with laboratory based condensation coefficients, the resulting errors in aerosol-cloud droplet number closure studies are still well above 10% and often around 20–30%, mostly due to overestimation (McFiggans et al., 2006).

In this study, a new spectral CPM, which solves explicitly the cloud microphysics of condensation, collision-coalescence, and lateral entrainment processes, was developed and is presented here. The new CPM was developed to be coupled to an existing rainfall microphysics column model describing the dynamic evolution of raindrop size distributions (bounce, collision-coalescence, and breakup mechanisms) (Prat and Barros, 2007b;Prat et al., 2012;Testik et al., 2011), but it will be made available publicly to the community in stand-alone form at GitHub. Previous research specifying cloud layers based on observed reflectivity profiles (Duan and Barros, 2017a;Wilson and Barros, 2014) showed that SFI can increase the intensity of rainfall by up to one order of magnitude in the SAM, consistent with ground observations. The purpose of the coupled process model is to investigate low-level precipitation enhancement induced by SFI among multilayer low-level clouds generally, and between locally initiated or propagating convective clouds and low-level boundary layer clouds in particular using local aerosol input. The model will be made avai

Observations during IPHEx (Integrated Precipitation and Hydrology Experiment; Barros et al., 2014) provide a great opportunity to apply this new CPM for the purpose of investigating ACI in an orographic context of the Southern Appalachian Mountains (SAM, see Fig. 1b). The focus here is on the evolution of cloud droplet spectra with height,that is the vertical microphysical structure of clouds. The numerical experiments described here aim to elucidate the quantitative impact of key ACI parameters (e.g., condensation coefficient, entrainment strength, hygroscopicity), as well as initial conditions (e.g. aerosol properties, thermodynamic conditions in the atmosphere) on early cloud development. Surface aerosol measurements sampled during IPHEx and sounding profiles from WRF simulations were used to initialize the parcel model. Model sensitivity experiments were conducted to test the model response to variations in major inputs and assumptions within their physically possible ranges. Vertical profiles of predicted cloud droplet concentrations and size spectra are compared with airborne measurements for a cumulus congestus case-study to elucidate determinant factors in the microphysical evolution of clouds at early stages in the SAM.

The manuscript is organized as follows. The mathematical formulation of the cloud parcel model is briefly described in Sect. 2. Section 3 presents the IPHEx measurements relevant for the modelling study. In Sect. 4, model sensitivity tests are conducted by varying key parameters of ACI within their physically-meaningful ranges and are compared against in situ observations to identify major contributors to cloud formation over the complex terrain of the SAM. Finally, a discussion of the main findings and a brief outlook of ongoing and future research are presented in Sect. 5.

## 2 Model description

To investigate the evolution of cloud droplet spectra originating from aerosol distributions of uniform chemical composition, a new cloud parcel model (hereafter DCPM, or Duke CPM for specificity) was developed to explicitly solve key cloud microphysical processes (see the last row of Table 1 for details). The model synthesizes well-established theory and



physical parameterizations in the literature. In particular, condensation and lateral homogeneous entrainment follow the basic formulations of Pruppacher and Klett (1997) and Seinfeld and Pandis (2006) albeit modified to incorporate the single parameter representation of aerosol hygroscopicity (Petters and Kreidenweis, 2007). The representation of collision-coalescence processes takes into account the variation of collision efficiencies with height (Pinsky et al., 2001), and the effects of turbulence

on drop collision efficiency as per Pinsky et al. (2008). The model discretizes the droplet spectra on a finite number of bins (*nbin*) using a discrete geometric volume-size distribution, spanning a large size range with fewer bins and very fine discretization in the small droplet sizes to improve computational efficiency (Kumar and Ramkrishna, 1996;Prat and Barros, 2007a). The characteristic single-particle volumes in adjacent bins are expressed as $v_{i+1} = V_{rat} v_i$, where $V_{rat}$ is a constant volume ratio (Jacobson, 2005). When condensation and coalescence are solved simultaneously, a traditional stationary (time-invariant)

grid structure often introduces artificial broadening of the droplet spectrum by reassigning droplets to fixed bins through interpolation, that is numerical diffusion (Cooper et al., 1997;Pinsky and Khain, 2002). To eliminate numerical diffusion artefacts, a moving grid structure is implemented so that an initial size distribution based on a fixed grid discretization can change with time according to the condensational growth. This approach allows particles in each bin to grow by condensation to their exact transient sizes without partitioning between adjacent size bins. Subsequently, collision and coalescence are

resolved on the moving bins that evolve from condensation. The DCPM predicts number and volume concentrations of cloud droplets and interstitial aerosols, liquid water content (LWC), effective drop radius, reflectivity and other moments of DSD. It also tracks thermodynamic conditions (e.g., supersaturation, temperature, pressure) of the rising air parcel. The flowchart in Fig. 2 graphically describes the key elements and linkages in the parcel model, including microphysical processes, and main inputs and outputs. A detailed description of the formulation of key processes in presented next. A glossary of symbols as well

as additional formulae are summarized in Appendix A. The performance of the DCPM was first evaluated by comparing its dependence on different parameters with the results from the numerical simulations reported by Ghan et al. (2011) as shown in Sect. S1 (see the supplementary material). Specifically, Figs S1-S6 demonstrate that the simulated maximum supersaturation and number fraction activated from the DCPM are in good agreement with the numerical solutions in Ghan et al. (2011) for a wide range of updraft velocities, aerosol number concentrations, geometric mean radii, geometric standard deviations,

hygroscopicity, and condensation coefficients.

**2.1 Condensation growth with entrainment**

The time variation of the parcel's temperature ($T$) can be written as

$$-\frac{dT}{dt} = \frac{gV}{c_p} + \frac{L}{c_p}\frac{dw_v}{dt} + \mu\left[\frac{L}{c_p}(w_v - w_v') + (T - T')\right]V \tag{1}$$

where the first two terms on the right-hand side represent adiabatically cooling of a rising parcel and the third term describes the modulation by entraining ambient dry air. The vertical profiles of ambient temperature ($T'$) and water vapour mixing ratio

($w_v'$) can be interpolated from input sounding data from atmospheric model simulations or radiosonde observations.
The change of the water vapour mixing ratio ($w_v$) in the parcel over time is described by



$$\frac{dw_v}{dt} = -\frac{dw_L}{dt} - \mu(w_v - w_v' + w_L)V \tag{2}$$

The change of the parcel's velocity ($V$) is given by

$$\frac{dV}{dt} = \frac{g}{1+\gamma}\left(\frac{T-T'}{T'} - w_L\right) - \frac{\mu}{1+\gamma}V^2 \tag{3}$$

where $\gamma \approx 0.5$ to include the effect of induced mass acceleration introduced by Turner (1963).

Due to significant uncertainties and complexities of entrainment and turbulent mixing (Khain et al., 2000), only lateral entrainment that mixes in ambient air instantaneously and is homogeneous in the parcel is considered in the DCPM. Based on observations from McCarthy (1974), the entrainment rate ($\mu$) is represented by an empirical relationship that describes the influx of air and ambient particles into the parcel as varying inversely with cloud radius. To describe the lateral entrainment, the bubble model (Scorer and Ludlam, 1953) and the jet model (Morton, 1957) are both incorporated in the parcel model.

For the bubble model, the change of the radius of a thermal bubble ($R_B$) over time is given as

$$\frac{dlnR_B}{dt} = \frac{1}{3}\left(\mu_B V - \frac{dln\rho_a}{dt}\right) \tag{4}$$

where $\mu_B = C_B/R_B$ and $C_B \approx 0.6$ (McCarthy, 1974).

For the jet model, the time variation of the radius of a jet plume ($R_J$) is expressed by

$$\frac{dlnR_J}{dt} = \frac{1}{2}\left(\mu_J V - \frac{dln\rho_a}{dt} - \frac{dlnV}{dt}\right) \tag{5}$$

where $\mu_J = C_J/R_J$ and $C_J \approx 0.2$ (Squires and Turner, 1962).

The condensational growth rate of droplets in the $i^{th}$ bin ($i = 1, 2,..., nbin$) is represented as

$$\frac{dr_i}{dt} = \frac{G}{r_i}(S - S_{eq}) \tag{6}$$

where droplet growth via condensation is driven by the difference between the ambient supersaturation ($S$) and the droplet equilibrium supersaturation ($S_{eq}$, see Eq. A4 in Appendix A). The growth coefficient (G) depends on the physicochemical properties of aerosols (see Eq. A1 in Appendix A).

Assuming $S \ll 1$, then $(1+S) \approx 1$, and the time variation of the supersaturation in the parcel can be expressed as

$$\frac{dS}{dt} = \alpha V - \gamma\left(\frac{dw_L}{dt} + \mu V w_L\right) + \mu V\left[\frac{LM_w}{RT^2}(T-T') - \frac{pM_a}{e_s M_w}(w_v - w_v')\right] \tag{7}$$

where $\alpha$ and $\gamma$ depend on temperature and pressure (see Eq. A5 and A6 in Appendix A).

During the parcel's ascent, entrainment mixes out cloud droplets and interstitial aerosols inside the parcel and brings in dry air and aerosol particles from the environment. Entrained aerosols are exposed to supersaturated conditions in the parcel. Some of them become activated and continuously grow into cloud droplets. The rate of change in droplet number in the $i^{th}$ bin ($i = 1, 2,..., nbin$) due to entrainment is

$$\left(\frac{dN_i}{dt}\right)_{ent} = -\mu V(N_i - N_i') \tag{8}$$




where $N'(z)$ is the number concentration of ambient aerosol particles (i.e. outside the cloud) at altitude z.

The rate of change in liquid water mixing ratio ($w_L$) in the parcel is calculated as follows

$$\frac{dw_L}{dt} = \frac{4\pi\rho_w}{3\rho_a} \sum_{i=1}^{nbin} \left( 3N_i r_i^2 \frac{dr_i}{dt} + r_i^3 \frac{dN_i}{dt} \right) \tag{9}$$

## 2.2 Collision-coalescence growth

To describe droplet growth by collision-coalescence process, the stochastic collection equation (SCE) that solves for the time
rate of change in the number concentration is written following Hu and Srivastava (1995)

$$\frac{\partial N(v)}{\partial t} = \frac{1}{2} \int_0^v N(v-v',t)N(v',t)C(v-v',v')dv' - N(v,t)\int_0^\infty N(v',t)C(v,v')dv' \tag{10}$$

where the first integral on the right-hand side of the equation describes the production of droplets of volume $v$ resulting from coalescence of smaller drops, and the second integral accounts for the removal of droplets of volume $v$ due to coalescence with other droplets. The continuous SCE is discretized and numerically solved by a linear flux method as outlined by Bott (1998). This method is mass conservative, introduces minimal numerical diffusion, and is highly computationally efficient (Kerkweg
et al., 2003;Pinsky and Khain, 2002). As noted before, the collision-coalescence process is calculated on a moving grid with bins modified by condensational growth at each time step.

For two colliding drops of volume of $v$ and $v'$, the coalescence kernel $C(v, v')$ in Eq. (10) is computed as the product of the gravitational collision kernel $K(v, v')$ and the coalescence efficiency $E_{coal}(v, v')$,

$$C(v,v') = K(v,v')E_{coal}(v,v') \tag{11}$$

$$K(v,v') = (9\pi/16)^{1/3}\left(v^{1/3} + v'^{1/3}\right)^2 |V - V'|E_{coll}(v,v') \tag{12}$$

where $V$ ($V'$) is the terminal velocity of drop volume $v$ ($v'$) and $E_{coll}(v, v')$ is the corresponding collision efficiency.
The terminal velocity of cloud drops is estimated following Beard (1976) in three ranges of the particle diameter (0.5 μm–19 μm, 19 μm–1.07 mm, 1.07 mm–7 mm). Another approximation by Best (1950) is also available as an option in the model. The table of drop-drop collision efficiencies at 1-μm resolution developed by Pinsky at al. (2001) is used for $E_{coll}$. This table was created based on simulations of hydrodynamic droplet interactions over a broad range of droplet radii (1–300 μm), including collisions among small cloud droplets as well as between small cloud droplets and small raindrops. Moreover, $E_{coll}$
was derived at three pressure levels of 1,000-, 750-, and 500-mb and can be interpolated at each level of a rising cloud parcel, thus taking the increase of $E_{coll}$ with height into account. Turbulence can significantly enhance collision rates especially for small droplets (below 10 μm in radii) as it increases swept volumes and collision efficiencies, and influences the collision kernels and droplet clustering (Khain and Pinsky, 1997;Pinsky et al., 1999;Pinsky et al., 2000). Considering different turbulent intensities for typical stratiform, cumulus, and cumulonimbus clouds, detailed tables of collision kernels and efficiencies in
turbulent flow, created by Pinsky et al. (2008) for cloud droplets with radii below 21 μm, are also incorporated in the model. $E_{coal}$ is parameterized following Seifert et al. (2005), who applied Beard and Ochs (1995) for small raindrops ($d_S < 300$ μm),



Low and List (1982) for large raindrops ($d_S$ > 600 µm), and used an interpolation formula for intermediate drops (300 µm < $d_S$ < 600 µm) where $d_S$ is the diameter of the small droplet. A simpler and faster option suggested by Beard and Ochs (1984) is also available in the model.

## 2.3 Numerical formulation

5        The equations in Sect. 2.1 constitute a stiff system of non-linear, first-order ordinary differential equations and involve state variables at very different scales. For the numerical integration of condensation growth, a fifth-order Runge-Kutta scheme with Cash-Karp parameters (Cash and Karp, 1990) using adaptive time steps (Press et al., 2007) is employed. At each time step, the error is estimated using the fourth-order and the fifth-order Runge-Kutta methods. Because dependent variables differ by several orders of magnitude, a fractional error ($\varepsilon$) is defined to scale the error estimate by the magnitude of each variable.

Specifically, the step size is adaptively selected to satisfy a fractional tolerance of $10^{-7}$ for all variables. The initial time step to calculate condensational growth is $5 \times 10^{-4}$ s. The maximum time step is set as $10^{-3}$ s to ensure the diffusional growth of drops is precisely simulated and non-activated particles reach equilibrium with the parcel supersaturation at each time step. For the collision-coalescence processes in Sect. 2.2, a simple Euler method is applied to integrate forward in time. The flux method for solving the discrete SCE was demonstrated to be numerically stable for various grid structures and integration time steps

when the positive definiteness is maintained (Bott, 1998). Thus, a time increment of 0.2s is chosen to assure that the available mass in each bin is much larger than the change of mass in the bin during the redistribution of the mass at one time step. Relying on separate numerical integration methods for calculating condensation and collision-coalescence allows us to either include or exclude each process easily to examine its role individually in cloud formation.

## 3 IPHEx data

The intense observing period (IOP) of the IPHEx field campaign took place during 01 May–15 June, 2014. The study region was centred on the SAM extending to the nearby Piedmont and Coastal Plain regions of North Carolina (see maps in Fig. 1). IPHEx was one of the ground validation campaigns after the launch of NASA's Global Precipitation Mission (GPM) core satellite, and details about this campaign can be found in the science plan (Barros et al. 2014). Surface measurements in the inner region of the SAM were conducted in the Pigeon River Basin (PRB, Fig. 1b) including a dense network of raingauges

and disdrometers. During the IPHEx IOP, measurements of aerosol concentrations and size distributions ranging from 0.01 to 10 µm were collected at the ground level in the Maggie Valley (MV), a tributary of the Pigeon River. Collocated with aerosol instruments at the MV supersite, the ACHIEVE (Aerosol-Cloud-Humidity Interaction Exploring & Validating Enterprise) platform was also deployed, equipped with W-band (94 GHz) and X-band (10.4 GHz) radars, a ceilometer, and a microwave radiometer. Two aircraft were dedicated to the IPHEx campaign. The NASA ER-2 carried multi-frequency radars (e.g., a dual-

frequency Ka-/Ku-, W-, X-band) and radiometers, and functioned as the GPM core-satellite sampling simulator from high altitude. The University of North Dakota (UND) Citation aircraft was instrumented to characterize the microphysics and





dynamical properties of clouds, including LWC and DSDs from cloud to rainfall drop sizes. Therefore, this data set offers a great opportunity to investigate ACI tied to warm season moist processes in complex terrain. Detailed description of the specific measurements relevant to this study is provided below and in Sect. S2.

### 3.1 Surface measurements

5       Aerosol observations were carried out at the MV supersite (marked as the yellow star in Fig. 1b) in the inner mountain region during the IPHEx IOP. The elevation of the MV site is 925 m mean sea level (MSL). This data set provides a clear characterization of the size distribution and hygroscopicity of surface aerosols in this inner mountain valley, which was not available previously. Nominal dry aerosol size distributions at the surface were measured by a scanning mobility particle counter system (SMPS) for particles from 0.01 to 0.5 μm in diameter, and a passive cavity aerosol spectrometer (PCASP; manufactured by Droplet Measurement Technologies, Inc., Boulder, CO, USA) for particle diameters in the size range of 0.1–10 μm. The SMPS consists of an electrostatic classifier (TSI Inc. 3081) and a condensation particle counter (CPC, TSI 3771). Note that the relative humidity (RH) of the differential mobility analyser (DMA) column is well controlled and the average RH (± one standard deviation) of the sheath and sample flows are 2.0±0.8% and 3.2±0.5%, respectively. In addition, a co-located ambient CPC (TSI 3772), which measures aerosol particles greater than 10 nm without resolving their size distributions, shows very close agreement with the SMPS measurements with regard to total number concentrations of aerosol particles ($N_{CN}$). Size-resolved CCN concentrations ($N_{CCN}$) were sampled by a single column CCN counter (manufactured by Droplet Measurement Technologies, Inc., Boulder, CO, USA) that was operated in parallel to the SMPS-CPC. The CCN instrument cycles through 6 levels of supersaturation (S) in the range of 0.09–0.51%. At a given S level, each CCN measurement cycle took approximately 8 min, corresponding to one SMPS-scan and buffer time to adjust supersaturation. On average 178 measurement cycles were collected daily during the IPHEx IOP, except for occasional interruptions due to instrument maintenance. CN and CCN distributions were inverted as described previously (Nguyen et al., 2014;Petters and Petters, 2016). Supersaturation was calibrated using dried ammonium sulfate and a water activity model (Christensen and Petters, 2012;Petters and Petters, 2016). The midpoint activation diameter ($D_{50}$) is derived from the inverted CN and CCN distributions (Petters et al., 2009). The hygroscopicity parameter ($\kappa$) is obtained from $D_{50}$ and instrument supersaturation (Petters and Kreidenweis, 2007). Detailed time series and diurnal cycles of CN and CCN measurements are presented in in Supplementary Sect. S2 (Figs. S7 – S9). The data show that the average total number concentration (± one standard deviation) of dry aerosol particles is 2,487±1,239 cm$^{-3}$ for particles with diameters between 0.01 to 0.5 μm, and 1,106±427 cm$^{-3}$ for particles with diameters between 0.1 and 10 μm in diameter. No significant diurnal variability in number concentration and hygroscopicity were observed. In addition, a co-located Vaisala weather station (WXT520) was continuously recording local meteorological conditions (e.g., wind speed, wind direction, relative humidity, temperature, and pressure) at 1-s interval. Diurnal cycles of these local meteorological variables during the IPHEx IOP are displayed in Fig. S10. The average meteorological conditions at the sampling site are 0.8±0.6 m s$^{-1}$ in wind speed, 172±115° in wind direction, 77±18% in relative humidity, 19±4 °C in ambient temperature (arithmetic mean ± one standard deviation).



## 3.2 Aircraft measurements

Airborne observations from the UND Citation, equipped with meteorological (e.g., temperature, pressure, humidity) sensors and microphysical instruments, are used in this study (Poellot, 2015). Vertical velocity was obtained from a gust probe and bulk LWC values were retrieved from two hot-wire probes (a King-type probe and a Nevzorov probe). Size-resolved concentrations were measured from three optical probes, covering droplet diameter from 50 μm to 3 cm: a PMS two-dimensional cloud (2D-C) probe, a SPEC two-dimensional stereo (2D-S) probe, and a SPEC high volume precipitation spectrometer 3 (HVPS-3) probe. The cloud droplet probe (CDP) measures cloud drop concentrations and size distributions for small particles with diameters from 2 to 50 μm in 30 bin sizes. The droplet sizes are determined by measuring the forward scattering intensity when droplets transit the sample area of the CDP. Coincidence errors have been found to cause CDP measurement artefacts, which tend to underestimate droplet concentrations and broaden droplet spectra. This type of error occurs when two or more droplets pass through the CDP laser beam simultaneously, and is highly dependent on droplet concentrations (Lance et al., 2010). Correction of the CDP observations is described in Sect. S2.2 (Fig. S11). The corrected CDP measured cloud droplet distributions were used in this study mainly to compare with model simulated cloud microphysics. The corrected CDP spectra slightly shift the measurement to smaller drop sizes (not shown here), thus providing confidence in the performance of the CDP probe during the IPHEx campaign.

## 3.3 IPHEx case-study: 12 June 2014

On 12 June 2014, the W-band radar observations at MV (see Fig. S12) indicate the formation of cumulus congestus clouds before 12:30 local time (LT) and further growth into cumulonimbus clouds. Near the MV site, a coordinated aircraft mission of both the UND Citation and NASR ER-2 was conducted from 12:14 to 15:51 LT on 12 June. Cloud droplet concentrations and size distributions were sampled by conducting successively higher constant-altitude flight transects through clouds. Droplet spectra were sampled at 1-Hz resolution (corresponding to approximately 90 m in flight distance) by the CDP and coincidence errors were taken into account by applying the correction as described in Sect. S2. In particular, the lowest horizontal leg (see the flight track in Fig. 3a, altitude around 2,770–2,800 m MSL) through the cloud is investigated to avoid the influence of substantial mixing in the upper portion of the cloud, which is not treated in the DCPM currently. The flight period of the first horizontal leg (~ 2,800 m MSL) is from 12:17 to 12:28 LT (See Fig. S13a). In rising updrafts, in-cloud samples (white plus signs in Fig. 6a and green crosses in Fig. S13) are defined with a minimum LWC of 0.25 g m⁻³ from the CDP. Along the first leg, three cloudy regions are identified near the eastern ridges (ER, highlighted in the blue dashed box in Fig. 3), over the inner valley region (IC, highlighted by the blue circle), and near the Eastern Cherokee Reservation (ECR, highlighted in the blue dashed box) and the corresponding measurements of in-cloud samples are present in Fig S13. In these cloudy regions, strong updrafts, and higher values of cloud drop numbers and LWC from the CDP are evident as shown in Fig. S14. The drop number concentrations from the 2-DC probe (measuring hydrometeors with diameter between 105 μm and





mm) indicate negligible amount of precipitation-sized drops in these cloudy regions (Fig. S14d), indicating the sampling of cumulus congestus clouds development by the aircraft.

To further eliminate regions influenced by mixing and other unresolved mechanisms, cloud segments to perform the modeling study are carefully selected by screening the cloud droplet spectra observed by the CDP. Following criteria 2 and 3 listed in Conant et al. (2004), measurements with effective droplet diameter greater than 2.4 μm and geometric standard deviation less than 1.5 are used in the analysis. During the first cloud transect, only one in-cloud region (IC, circled in Fig. 3a) satisfies Conant et al.'s requirements with 11 cloudy samples (corresponding vertical velocities shown as blue bars in Fig. 3b) collected over approximately 1 km flight distance. Significant topographic heterogeneity (terrain transect indicated by the thick black line in Fig. 3b) can exert a considerable influence on cloud formation across this region. As shown in Figs. 3c and d, a pronounced variability in drop number distributions is manifest in the in-cloud samples clustered by low (0–1 m s$^{-1}$) and high (1–2 m s$^{-1}$) updrafts. As expected, the droplet spectra in stronger updrafts at the core (see Fig. 3d) have higher number concentrations and narrower size range compared to the samples at the edge of the cloud (see Fig. 3c). Moreover, droplet spectra measured within updraft core of two other cloudy regions in the inner SAM (highlighted in dashed light blue boxes in Fig. 3a) as well as IC are shown in the top panel of Fig. S15.

As aerosol size distributions are not resolved in the CPC measurements from UND Citation (see Fig. S15d), we resort to the surface sampling of aerosol concentrations at MV (marked as the black asterisk in Fig. 3a) as the input for modeling study at IC. Moderate vertical velocities measured in IC region (Fig. 3b) and analysis of the radar profiles at MV (Fig. S12) suggest that the early development phase of the cumulus congestus observed in the inner SAM was sampled by the aircraft on June 12 during IPHEx. We also should note that observed variations in vertical velocities and cloud droplet number concentrations over complex terrain are indicative of challenges in the application of parcel models as homogeneity is assumed for aerosol concentrations below cloud base and within the microstructure of the air parcel.

## 4 Modelling Experiments

### 4.1 Model initialization and reference simulation

Dry aerosol concentrations measured by the SMPS and PCASP at MV were averaged over the first 10 mins (averaging interval: 12:14 LT–12:24 LT) of the 12 June flight, and then merged into a single size distribution as shown in Fig. 4. The combined aerosol distribution at the surface is fitted by the superimposition of four lognormal functions using least-squares minimization. Table 2 summarizes parameters (total number concentration, geometric mean diameter, and geometric standard deviation) that characterize the four lognormal distributions. Notice that aerosol number concentrations below 0.03 μm are underestimated by the fitted cumulative distribution (cyan curve in Fig. 4). These particles in such small sizes mostly remain non-activated under the supersaturated conditions typical of the atmosphere, thus, underestimation of their concentrations does not affect cloud development in the model. The aerosol distribution is discretized into 1,000 bins, covering the size range of 0.01–10 μm. The bins are spaced geometrically with a volume ratio of 1.026. The bin grid at such a high resolution is sufficient



to precisely simulate the partitioning of growing droplets and interstitial aerosols in the parcel. It is also assumed that the aerosol is internally mixed so that the hygroscopicity does not vary with particle size. Thus, we prescribe a κ value of 0.14 for each aerosol bin, deriving from the average κ during the first 10 mins of the 12 June flight.

During the IPHEx IOP, daytime radiosondes were launched every 3-hours at Asheville, NC (red star in Fig. 1b). This location is on the eastern slopes of the SAM in the French-Broad valley outside of the inner mountain region far away from the targeted cloudy region (IC). In addition, the closest sounding (11 LT) was launched much earlier than the flight take-off time on 12 June 2014. To address the lack of sounding observations needed for CPM input, high-resolution (0.25-km grid size) WRF simulations were conducted to extract model soundings in the IC region (highlighted in Fig. 3a). Detailed configuration of the WRF model for these simulations (see Fig. 5a for nested grid domains) is described in Sect. S3. Upon inspection of model results 15 min prior to the flight time, WRF-simulated soundings from six columns in the IC region at 12:15 LT were averaged to estimate vertical profiles of ambient temperature and RH for the case-study as shown in Fig. 5b. The cloud base height (CBH) is chosen as the level where simulated RH is approximately 100%. As marked by the horizontal black line in Fig. 5b, CBH = 1,270 m AGL at 12:15 LT when the parcel is released from cloud base. The vertical distribution of simulated horizontal winds along the aircraft flight path is highly heterogeneous and anisotropic due to the complex 3D structure of winds in the complex terrain of the inner mountain region including shallow thermal upslope winds between the main valley and surrounding ridges and ridge-valley circulations with multiple orientations in lateral tributary valleys as illustrated by the supplementary animations SA1 (near surface), SA2 (at ridge level), SA3 (at CBH) and SA4 (along the first aircraft flight leg). The animations show southerly mesoscale horizontal transport above ridges, upslope flows along the topography in the inner region, and a mesoscale honeycomb like structure of weak to moderate updrafts and downdrafts with short-lived intensification linked to overturning processes across the entire region.

At the cloud base of IC, aerosol size distributions are estimated by assuming that total number concentrations at the surface decay exponentially with a scale height ($H_S$) of 1,000 m (representative of the effectiveness of the vertical venting mechanism), and geometric mean diameters and corresponding geometric standard deviations remain constant with height. The dry aerosol distribution at cloud base is calculated as the sum of four lognormal distributions with fitted parameters indicated in the last three columns of Table 2 and is taken as initial input to the model. The number concentration of entrained ambient aerosol particles (see Eq. 8) is also calculated based on the assumption that the initial aerosol distribution at the surface $N(0)$ decays exponentially with height: $N'(z)=N(0)exp(-z/H_S)$, where $z$ is the height above ground level (AGL) and $H_S$ is the scale height, depending on aerosol types (Kokhanovsky and de Leeuw, 2009).

The temperature excess of the air parcel over the environment is initialized as 1.0 K, and the initial pressure and RH of the parcel at cloud base adapt to cloud surroundings. As vertical velocities were not sampled at cloud base, the initial updraft velocity ($V_0$) is assumed to be uniformly distributed and equal to 0.5 m s$^{-1}$, consistent with vertical velocities observed by the W-band radar (see Fig. S12b) and simulated by the model around the same altitude (2.5 km MSL). Therefore, the air parcel is launched at cloud base with an initial parcel radius (R) of 500 m, an initial updraft of 0.5 m s$^{-1}$, and initial aerosol particles that are in equilibrium with the humid air at cloud base. When the parcel is rising, the lateral entrainment is treated as the bubble



model parameterization with the characteristic length scale R = 500 m (see Eq. 4 in Sect. 2.1). Ambient aerosol particles penetrate through lateral parcel boundaries and their number concentrations also decrease exponentially with height ($H_S$ = 1,000 m). The turbulent kinetic energy dissipation rate is chosen as 200 cm²s⁻³, typical of cumulus clouds at early stages. The parcel reaches cloud top when vertical velocity is near zero. Note that despite specified as stated above for the reference

simulation, sensitivity to parcel radius R, scale height $H_S$, and hygroscopicity κ will be explored in Sect. 4.2.

## 4.2 Parameter sensitivity analysis

In this section, sensitivity tests are conducted to assess changes in model simulations to variations in key inputs and assumptions. Test results are further compared with in-cloud observations from the aircraft to assess the role of individual state variables and processes for the cumulus congestus case-study on 12 June during IPHEx. Selected parameters are perturbed one

at a time while other assumptions and input parameters remain as specified in Sect. 4.1.

### 4.2.1 Condensation coefficient

Condensation plays a dominant role in the early stages of cloud formation and one key factor in this process is the condensation coefficient ($a_c$) that governs activation and condensational growth. A laboratory study by Chuang (2003) reported $a_c$ values ranging from $4 \times 10^{-5}$–1, and experimental values from field campaigns and from chamber studies of

individual droplet growth also differ over a wide range (0.007–0.1) as reviewed in Sect. 1. In this study, $a_c$ was made to vary in the range [0.001, 1.0] on the basis of Fountoukis and Nenes (2005). For the targeted in-cloud region (IC), Fig. 6 shows simulated profiles of updraft velocity, supersaturation, total CDNC, LWC, and their sensitivity to selected $a_c$ values in comparison with the airborne observations (denoted by black crosses). Measurements from the IC region along the lowest cloud transect (highlighted in the blue circle in Fig. 3a) are used to evaluate model performance, since no observations are

available in the upper unmixed cloudy areas to assess the entire vertical profiles simulated by the CPM. Only simulations with reasonable agreement with the observations are presented, thus results with $a_c$ from 0.06 to 1.0 are not shown here. Particles above 1 μm in diameter are considered cloud droplets and are included in the integration to calculate LWC. Note that ground elevations under the IC region vary from 928 m to 1,184 m MSL (see Fig. 3b), and the region is on a small hill in the middle of the valley and surrounded by much higher ridges (terrain elevation ~ 1,500 m MSL). Hereafter, aircraft measurements are

expressed as AGL to facilitate their comparisons with the model results.

As illustrated in Fig. 6a, large values of $a_c$ (> 0.01) have negligible influence on the vertical velocity profiles. Although it is apparent that $a_c$ has a significant impact on the simulated supersaturation profiles (Fig. 6b). The black crosses indicate the quasi-steady approximation of supersaturation ($S_{qs}$), calculated according to Eq (A8) (also Eq (3) in Pinsky et al. (2013)). We should keep in mind that large uncertainties are associated with the temperature measurements from the aircraft,

which are used to compute the $S_{qs}$. Low values of $a_c$ strongly inhibit the phase transfer of water vapour molecules onto aerosol particles (aerosol wetting), slowing the depletion of water vapor in the parcel, and thus substantially increasing maximum





supersaturation ($S_{max}$). Consequently, smaller aerosol particles with high concentrations are activated due to a higher $S_{max}$, resulting in a direct increase in cloud droplet numbers with lower values of $a_c$ (Fig. 6c). Overall, these results are in agreement with earlier studies (Nenes et al., 2002;Simmel et al., 2005) that investigated the dependence of cloud droplet number concentrations on the condensation coefficient. Moreover, Fig. 6c shows that the simulation with $a_c$ = 0.01 (green line)

captures well the observed drop concentrations between 1,500 m and 1,600 m AGL (highlighted in yellow shade), whereas a condensation coefficient that is one order of magnitude lower ($a_c$ = 0.002, blue line) yields better results for the observations above 1,600 m. As summarized in Table 3, the simulated CDNC for the region between 1,500 m and 1,600 m AGL along the hillslope (shaded in Fig. 6b, reference sub-region within IC) for $a_c$ = 0.01 attains an average CDNC of 354 cm$^{-3}$, which is only ~1.3% higher than the observed average between 1,500 m and 1,600 m (349.4 cm$^{-3}$). The corresponding LWC is also in

reasonable agreement with the range of observed values (Fig. 6d). The observed CDNC for the cluster between 1,600 m and 1,750 m (397.5 cm$^{-3}$) is underestimated by all simulations with different $a_c$ values and the averaged CDNC using a much lower condensation coefficient (0.002) is ~ 8% lower than the averaged observation. Inspection of Fig. 6c suggests that within IC there are two clusters of air parcels at different levels above ground. Interestingly, the cluster at lower elevation (Fig. 3b) is better reproduced using a lower condensation coefficient, whereas a higher condensation coefficient better captures the

reference region that includes the updraft core near the hilltop. Good agreement between the model results and airborne observations for the lower cluster provides more confidence in the conclusions about the sensitivity tests. Thus, the lower cluster is considered as the reference region within IC and will be the focus of this study.

       The sensitivity of predicted spectra at 1,500 m (in solid lines, Fig. 7a) to $a_c$ varying from 0.002 to 0.06 is very high. The observed spectrum (black dotted line) is the average from five individual CDP measurements (dotted lines in Figs. 3c and

d, also highlighted in the yellow shaded area in Fig. 3b) between 1,500 m and 1,600 m AGL (see Fig. 6d for their LWC in shade). Generally, spectra simulated with lower values of $a_c$ are broader with higher numbers of small droplets, while simulations with large values of $a_c$ yield narrower spectra shifted to larger droplet sizes. The differences in drop size range and spectra shape can be explained by inspecting the vertical profiles of the parcel supersaturation and $S_{eq}$ for six illustrative aerosol particle diameter ($D_{aero}$) depicted in Fig. S19. Growth by water vapour condensing on different sizes of cloud droplets

is determined by the difference between S and $S_{eq}$ (Eq. 6 in Sect. 2.1). At low S, small particles become interstitial aerosols, and their corresponding $S_{eq}$ remains in equilibrium with the parcel supersaturation (S - $S_{eq}$ = 0). At high S, as a result of low $a_c$ values, activation of small aerosols contributes to significant spectra broadening, produces larger CDNC, and shifts the droplet size distribution toward smaller diameters due to slower condensational growth. This is consistent with Warner (1969) who found that low condensation coefficients (< 0.05) were required to capture the observed dispersion of droplet spectra in

natural clouds, especially for small sizes (i.e. left-hand side of the spectra). Figure 7b displays the simulated droplet number distributions at different levels for $a_c$ = 0.01 in comparison with the individual droplet spectra measured by the CDP. The simulated spectra are representative of the evolution of cloud droplet distributions in one parcel at different cloud development stages. The observed spectrum at 1,559 m AGL (black dotted line) and its CDNC (357 cm$^{-3}$) and LWC (0.37 g m$^{-3}$) is chosen for the comparison as best agreement is achieved with the DCPM results. The results are also consistent with the



parameterizations of CDNC and cloud droplet spectra at different heights given the updraft velocity and the number of CCN that can be activated at moderate supersaturation levels developed by Kuba and Fujiyoshi (2006) using a cloud microphysical model. Simulated spectra at 1,500 m and 1,600 m altitude show very good agreement with the observed number concentration and drop size range. Below 1,600 m, a shift of the unimodal spectra to larger drop sizes suggests that the condensation process currently dominates the growth of cloud droplets. Larger drops above 1,700 m could be produced by coalescence growth, leading to the formation of a second mode at larger sizes in the upper portion of the cloud. For the analyses presented hereafter, we consider $a_c = 0.01$ together with other initial conditions as prescribed in Sect. 4.1, as the reference simulation (denoted by the grey line in the following figures).

Further examination using data from other cloud and precipitation probes suggests that concentrations of droplets larger than 30 µm in diameter are negligible during the first horizontal flight leg. Considering that droplets with diameters larger than 30–32 µm are required to trigger effective droplet collisions (Pinsky and Khain, 2002), we conclude that the collision-coalescence process is not important in the sampled IC region, and it is unlikely that it contributes to the wide bimodal spectra observed at early stages of cloud growth. It is noteworthy that small drops are absent in the simulated spectra, in contrast to the observed spectrum that exhibits a broad drop size range and two distinct modes (see Fig. 7b). One possible explanation is that the moving bin grid determined by the condensation process tends to widen the spectral gap between the growing droplets and non-activated aerosol particles in the ascending parcel. Thus, a geometric size distribution with 1,000 bins is utilized herein to further refine the discretization for small particle sizes. Another explanation relates to the uncertainties of the input sounding extracted from the WRF simulation. Even though ambient aerosols are entrained continuously through lateral boundaries, most of them remain as interstitial aerosol particles because the low supersaturation in the parcel is insufficient to enable activation (see Fig. 6b). The WRF sounding in Fig. 5b exhibits a lapse rate of -4.1 °C km$^{-1}$ from 1,270 m (CBH) to 2,200 m, corresponding to stable atmospheric conditions unfavourable for cloud development. To assess the impact of the environmental conditions on cloud growth, an additional model simulation was performed by altering lapse rate at lower levels (see Appendix B1). The results point out that uncertainties of the assumed environmental thermodynamic conditions (e.g., temperature) impose significant constraints in the vertical development of clouds, thus posing as a significant challenge in cloud modelling studies.

### 4.2.2 Entrainment strength

To access the influence of entrainment on cloud drop concentrations and LWC, different strengths of lateral entrainment are examined by altering the initial cloud parcel size R at the cloud base. Figure 8 displays the vertical profiles of total CDNC and LWC, and cloud droplet spectra formed at three altitudinal levels (1,500 m: solid line, 1,600 m: dotted line, and 1,700 m: dashed line) for simulations using different initial parcel radii as compared to the CDP observations in the IC region (denoted by black crosses in Figs. 8a and b and the black dotted line in Fig. 8c). Entrainment appears to have a dominant influence on the cloud vertical structure as small rising parcels associated with higher entrainment dissipate faster by intensive



mixing of dry ambient air through lateral cloud boundaries. Stronger entrainment strength results in a direct decrease in drop concentrations and LWC, while it has little influence on the droplet size range. The best agreement on droplet numbers is between the reference simulation (R = 500 m, $a_c$ = 0.01; grey line in Fig. 13a) and the reference sub-region within IC (between 1,500 m and 1,600 m AGL), whereas results for R = 1,500 m better captures the higher cluster of cloudy samples (above 1,600

m AGL). Recall that previously, when R was held constant the higher cluster is better reproduced using $a_c$ values one order of magnitude smaller than the reference value. Thus, the sensitivity analysis does suggest there is a trade-off with weaker entrainment for a higher condensation coefficient (R = 1500 m and $a_c$ = 0.01, the orange line in Fig. 8a) when other parameters in the reference simulation remain the same.

       Given R = 500 m, an additional test was conducted using the jet model parametrization of lateral entrainment (Eq. 5

in Sect. 2.1). The comparison of two entrainment parameterizations indicates that the bubble model (the grey line) has stronger entrainment strength than the jet model (red line) given the same initial parcel size (500 m). Nevertheless, continuous increases in simulated LWC in the upper portion of the cloud (see Fig. 8b) for both parameterizations are unrealistic in real clouds (Paluch, 1979). This problem can be likely ascribed, at least in part, to the uncertainties in the environmental conditions associated with the WRF sounding. As noted in Fig. B1, decreases in LWC are manifest at the upper portion of the cloud, as

indicated in the simulations with modified sounding inputs. The lack of sufficient mixing with dry ambient air near cloud top is an inherent deficiency in the simple parameterization of lateral homogenous entrainment, assuming decreasing entrainment strength with height, but this assumption does not significantly affect our conclusions for in-cloud regions below cloud top.

### 4.2.3 Initial aerosol concentration

       The initial aerosol concentration at cloud base can also have significant effects on cloud development. Because aerosol

size distributions were not sampled by the aircraft during IPHEx, they are estimated by extrapolating surface aerosol number concentrations according to an exponential decay with a given scale height ($H_S$). To probe and characterize the dependence of droplet formation on aerosol concentrations available at cloud base, sensitivity to $H_S$ was explored by varying its values from 800 m to 1,200 m. Figure 9 shows the simulated profiles of the total CDNC and LWC, and cloud droplet spectra formed at three altitudinal levels (1,500 m: solid line, 1,600 m: dotted line, and 1,700 m: dashed line). It is not surprising that aerosol

concentrations at cloud base have a substantial influence on the resulting droplet concentrations. Higher aerosol concentrations, inferred from larger $H_S$ lead to larger drop numbers with smaller average droplet sizes, which is known as the first indirect effect of aerosols (Twomey, 1977). Yet, here, LWC appears insensitive to the initial aerosol concentration as it is limited by moisture content available in the parcel. The value of $H_S$ = 1,000 m yields the best agreement in CDNC between the DCPM simulations and the average droplet spectra observed by the CDP (black dotted line in Fig. 9c, see reference sub-region within

IC shaded in Fig. 3b), which lies within the typical $H_S$ range (550–1,100 m) of aerosol number concentration measurements for remote continental type (Jaenicke, 1993).





Because of the uncertainty in the characterization of environmental conditions due to the lack of sounding observations and the complexity of 3D circulations in the inner mountain region, an additional set of CPM simulations were conducted assuming a well-developed and well-mixed CBL, and thus uniform distribution of dry aerosol concentrations below CBH. This enables contrasting the results using the well-mixed CBL and the vertical venting mechanism to pump low level

aerosol to the atmosphere above the mountain ridges. The modelling results are present and discussed in Sect. S4. The surface aerosol concentration at MV (see Fig. 4) is used as model input at cloud base, and other input parameters remain as specified in Sect. 4.1. Although there is good agreement in CDNC between simulations with surface aerosols at cloud base and the airborne observations as expected using a conservative CPM, there are large discrepancies between the observed and simulated cloud droplet spectra with respect to spectral width, peak diameter and peak concentration number above CBH. As noted in

Fig. S15d, aerosols in the atmosphere exhibit a significant space-time variability especially in regions of complex terrain with heterogeneous mixing by different ventilation processes in addition to the possibility of remote transport, all of which can contribute to the diverse cloud droplet spectra observed across the cloud transect (see Figs. S15a-c). This cannot be captured by the current model simulations that assume a homogenous aerosol distribution at cloud base.

### 4.2.4 Hygroscopicity

Another key element in the condensation process is the hygroscopic property that governs the influence of aerosol chemical composition on CCN activity. To account for its temporal variability observed during IPHEx, a κ value varying from 0.1–0.4 (within the typical range measured at the surface site, see Figs. S8a and S9c) is applied uniformly for all particle sizes. As noted from Fig. 10, simulated profiles of total CDNC and LWC exhibit a weak dependence on the hygroscopicity with only a slightly increase in total CDNC with more hygroscopic aerosols. Predicted droplet spectra at three altitudinal levels

(1,500 m: solid line, 1,600 m: dotted line, and 1,700 m: dashed line) also show little sensitivity to the variations in κ. As discussed in Sect. S2, hygroscopic properties of aerosols have been found to vary with particle sizes. Potential uncertainties might remain by assuming a constant κ, but its variation with droplet sizes is not addressed in the current study. We should also note the hygroscopicity derived from surface measurements may not be representative for aerosols beneath the cloud (Pringle et al., 2010). However, the vertical variability of aerosol hygroscopicity is not taken into account in this study.

### 4.2.5 Summary of sensitivity analysis

Under realistic assumptions, the total number concentration and size distributions from the airborne observations are captured well by the reference simulation. Sensitivity tests by changing $a_c$ in the range of 0.001–1.0 suggest that the predicted CDNC, LWC, and thermodynamic conditions are highly dependent on the condensation coefficient. At early stages of cloud development, the condensation coefficient plays a key role in the simulated spectra width and shape that increases in $a_c$ lead

to a shift towards larger droplet sizes and narrower spectra widths. Entrainment has a substantial impact on the cloud depth, droplet numbers, and LWC, whereas initial aerosol concentrations have a strong effect on number concentrations and size



distributions of cloud droplets, but induce little effects on LWC. Hygroscopicity has negligible influence on simulated total CDNC and LWC. Additional tests regarding the sounding inputs and initial updraft velocity were conducted and discussed in Appendix B.

Due to the limited dataset from the campaign, a specific set of initial conditions are inferred from surface and airborne observations and reasonable assumptions are made based on the literature and WRF model results. It is important to keep in mind the uncertainties associated with the determination of CBH, which is estimated from the WRF model simulations as concurrent soundings are not available during IPHEx. If the CBH is lifted by 100 m, simulations using different $a_c$ values (0.002 – 0.06) are in better agreement with the airborne measurements of LWC. The CDNC in the reference region (yellow shade, Fig. 6c) is captured better with a higher $a_c$ value (0.015) but narrower spectra results are associated with increasing $a_c$ values, inconsistent with the observed spectra (not shown here). These caveats highlight the need for comprehensive concerted observations of end-to-end processes in future field campaigns.

Based on the sensitivity tests, model simulations using a relatively low value of $a_c$ (0.01) exhibit CDNC and spectra consistent with the cloud spectra observed in the inner region of the SAM for early development of cumulus congestus on 12 June. Assuming adiabatic conditions (i.e. in the absence of entrainment) and for aircraft observations near cloud base (discussed in Sect. 1), higher values of the condensation coefficient  have been specified in previous studies such as $a_c = 0.06$ for wam cumulus during CRYSTAL-FACE (Conant et al., 2004), $a_c = 0.042$ for stratocumulus during Coastal Stratocumulus Imposed Pertubation Experiment (CSTRIPE, Meskhidze et al., 2005), and $a_c = 0.06$ for cumuliform and stratiform clouds during ICARTT (Fountoukis et al., 2007), consistent with the modal values in Shaw and Lamb (1999). In this work, entrainment is included and comparisons against the observations are performed several hundred meters above cloud base. Exploratory simulations assuming a higher aerosol number concentration at cloud base ($H_S$ = 1,200 m, Fig. B3b) show a highly nonlinear response to changes in $a_c$ and R with the best agreement in CDNC being achieved with higher $a_c$ values (0.03 and 0.06) and weak entrainment (R = 1500 m) consistent with the nonlinear trade-offs between entrainment (stronger, R = 500 m) and condensation (lower, $a_c = 0.01$) for the reference simulation as discussed in Sect. 4.2.2. Nevertheless, the corresponding spectra simulated with higher $a_c$ values show larger discrepancies in spectral width and shape against the observations within IC (not shown here), and thus predictions of inferior skill with regard to cloud vertical development.

One possible explanation for the lower condensation coefficients in IPHEx could be the presence of organic film-forming compounds (FFCs) on the surface of natural aerosol particles (Feingold and Chuang, 2002). Organic films can strongly impede the uptake of moisture by atmospheric aerosol particles, thus reducing the value of $a_c$ (Gill et al., 1983;Mozurkewich, 1986). Nenes et al. (2002) conducted a parcel model study to investigate the impact of aerosol coating with organic FFCs using a constant $a_c$ and concluded that the initial condensational growth is impeded, leading to higher supersaturations in the parcel, and increasing the cloud droplet number by a substantial amount due to a higher number of activated CCN. Without the characterization of the organic speciation in this campaign, the presence of organic coating on local aerosols remains an open question. Finally, results from laboratory experiments of direct contact condensation on aerosols in cloud chambers with horizontal or vertical moist flows, point to $a_c$ values around 0.01 (Garnier et al., 1987;Hagen et al., 1989) in contrast with the





most probable value (0.06) found in the levitation cell by Shaw and Lamb (1999). Fukuta and Myers (2007) demonstrated that the lower values of $a_c$ could be explained using diffusion-kinetic theory explicitly accounting for feedbacks between latent heat and temperature in the boundary-layer of growing droplets.

## 5 Summary and discussion

5       In this study, a new entraining cloud parcel model (DCPM) with explicit bin microphyscis is presented and applied to explore the vertical structure of cloud development, and to investigate dominant factors in determining the microphysical development of clouds in the complex terrain of the inner SAM. Model evaluation specifically focuses on the development of a mid-day cumulus congestus case on 12 June 2014 when aircraft measurements are available during the IPHEx campaign. Although this flight sampled three distinct cloud regions along the lowest aircraft transect, the IC region and the MV supersite

are closely located in the inner valley region of the SAM, and thus a detailed modelling study could be conducted leveraging ground-based aerosol measurements and W-band radar profiles available at MV to inform model initialization. Because measurements of key input parameters are not available from IPHEx, or cannot be resolved by current sampling techniques, a specific set of initial conditions and parameters was inferred from MV observations and the literature. Sensitivity experiments were conducted to examine the model dependence on key inputs and assumptions within their possible ranges. Albeit a large

variability in cloud microphysical properties was observed at sub-km scale (~ 90 m is the spatial averaging resolution of the measurements along the flight track) over the complex terrain of the inner SAM even within IC, modelling results for the reference simulation achieved good agreement with the observed CDNC, LWC, and droplet size spectra collected roughly 300–500 m above cloud base in the cloud updraft core, which lends credence to the findings of sensitivity tests.

       In the framework of the physically based cloud parcel model, sensitivity of the simulated cloud microphysical

characteristics to variations in key parameters was investigated within the context of in situ measurements. Results from sensitivity tests show that condensation coefficient exerts a profound influence on the droplet concentration, size distribution, LWC, and thermodynamic conditions inside the parcel, with a decrease in $a_c$ leading to an increase in cloud droplet number, a broader droplet spectrum, and a higher maximum supersaturation near cloud base. The case-study during IPHEx reveals that the observed cloud features in the inner mountain region of the SAM are better captured by a low value of $a_c$ (0.01) and strong

entrainment strength corresponding to R = 500m using the bubble parameterization of entrainment processes. As expected, entrainment is found to be a major process controlling the vertical structure, CDNC, and LWC of the cloud. Further, it was shown that with other input parameters remain the same as reference simulation conditions, there is a trade-off between the CDNC sensitivity to entrainment strength and the condensation coefficient: strong entrainment (meaning the characteristic scale R in the bubble parameterization is small) is compensated by lower $a_c$ values, and vice-versa. This competitive

interference explains higher values found in previous aerosol-CDNC closure studies assuming adiabatic cloud conditions (zero entrainment) in the CPMs. Initial aerosol concentrations at cloud base also have a large impact on droplet numbers but negligible influence on LWC. Hygroscopicity, however, has little effect on the simulated profiles of CDNC and LWC.



Nevertheless, analysis of the effect of the interdependence of initial aerosol concentration, condensation coefficient and entrainment strength on the CDNC revealed ambiguous behavior that could only be resolved by assessing the properties of the simulated droplet spectra (shape, range) against the aircraft measurements at different altitudes throughout the clouds (i.e., well above cloud base). Overall, these findings provide insight into key parameters of ACI processes in this region.

Finally, data and model limitations should be acknowledged. Regarding data limitations to constrain and force the CPM, reasonable assumptions were made based on the literature, surface and airborne observations from IPHEx, and WRF model simulations due to the lack of near cloud-base measurements and sounding, as pointed out earlier. Next, it is important to recognize the limitations of the lateral homogeneous entrainment employed in the model. Its concept is based on a simple assumption that entrained aerosols are mixed instantly across the parcel, which neglects the inhomogeneous supersaturation

and microphysical structure inside the cloud associated with discrete entrainment events on different spatial scales (Baker et al., 1980;Khain et al., 2000). Realistic entrainment and mixing with cloud surroundings have been found to contribute significantly to droplet spectrum broadening. Turbulent mixing (Krueger et al., 1997) can break down entrained blobs of air into smaller scales and subsequently form small adjacent regions with uniform properties on account of molecular diffusion, thus leading to considerable spectrum broadening. In addition, the parameterization of entrainment through lateral boundaries

neglects entrainment with dry air at cloud top that is expected to be an important element to cloud vertical development (Telford et al., 1984). Downdrafts induced by the penetration of dry air at cloud top can sink and mix with updrafts, effectively diluting number concentrations and broadening droplet spectra in clouds (Telford and Chai, 1980). Another limitation in the current approach is the assumption of uniform hygroscopic properties for all particle sizes. In reality, the aerosol distribution is an aggregate of particles with different physicochemical properties, including different shapes, solubility, and chemical species

(Kreidenweis et al., 2003;Nenes et al., 2002). Even if specified initial aerosol characteristics were to capture the variation of $\kappa$ with size, how to track the evolution of $\kappa$ as particles among different bins undergo coalescence and breakup remains a challenge. Nevertheless, the sensitivity analysis indicates that the cloud droplet growth is generally insensitive to hygroscopicity (Sect. 4.2.4), thus the constant $\kappa$ value used in this study does not significantly affect our modelling results.

For unstable cloud layers, complexity of in-cloud vertical velocity fields with localized areas of much stronger

updrafts has been found to support the formation of wide bimodal spectra in cumulus clouds due to in-cloud nucleation of new droplets from interstitial aerosols when the parcel supersaturation higher up in the cloud exceeds the cloud base maximum (Pinsky and Khain, 2002). As a result, this mechanism can lead to the formation of a secondary mode of small droplets in individual spectra, different from our observed spectra with a second mode centred at a larger droplet size (Figs. 3 and 7). In this study, however, supersaturation does not increase above the cloud base maximum under the conditions of the original and

modified model environments, likely attributed to the ambiguities in the sounding input from WRF, even if direct aircraft measurements albeit highly uncertain suggest otherwise.

The present study underlines the importance of the relationhsip between entrainment processes that determine the local- (microscale) and cloud-scale thermodynamic environment around individual particles, and the aerosol condensation coefficient that measures the effectiveness of condensation processes in the same thermodynamic environment. Given the



multiscale thermodynamic structure of clouds, these interactions suggest that realistically the condensation coefficients in the natural environment are transient and spatially variable. Therefore, further research to arrive at representative ensemble estimates are necessary to reduce the associated uncertainties of the aerosol indirect effect. In the present study, the local sensitivity of selected model parameters are assessed individually over certain ranges based on IPHEx data and the literature,

5 which ignores non-linear interactions among ACI modeling parameters as discussed above. Future work will focus on exploring the sensitivity of the DCPM in a multi-dimentional parameter space to quantify multiple parameter interactions (Gebremichael and Barros, 2006;Yildiz and Barros, 2007) on ACI processes using the fractorial design method (Box et al., 1978).

**Appendix A**

10 *Glossary of Symbols*

$a_c$ — condensation coefficient

$a_T$ — thermal accommodation coefficient

$c_p$ — specific heat of dry air

$D_v, D_v'$ — diffusivity of water vapor in air, modified diffusivity of water vapor in air

$e_s$ — saturation vapor pressure

$g$ — gravitational constant

$G$ — growth coefficient

$H_S$ — scale height

$k_a, k_a'$ — thermal conductivity of air, modified thermal conductivity of air

$L$ — latent heat of evaporation

$M_a, M_w$ — molecular weight of dry air, of water

$N, N'$ — number concentration of cloud droplets, of ambient aerosol particles

$p$ — pressure

$r, r_c$ — radius of cloud droplet, of dry aerosol particle

$R$ — universal gas constant

$R_a$ — specific gas constant for moist air

$R_v$ — Specific gas constant for water vapor

$R_B, R_J$ — radius of air bubble, of convective jet

$S$ — supersaturation





| | |
|---|---|
| $S_{eq}$ | droplet equilibrium supersaturation |
| $T\,(T')$ | temperature of air parcel (ambient air) |
| $V$ | parcel updraft velocity |
| $v,\,v'$ | droplet volumes |
| $w_L$ | mixing ratio of liquid water in parcel |
| $w_v\,(w_v')$ | mixing ratio of water vapor in parcel (in environment) |
| $\kappa$ | hygroscopicity parameter |
| $\mu$ | entrainment rate |
| $\rho_a,\,\rho_w$ | density of dry air, of water |
| $\sigma_w$ | droplet surface tension |

*Additional Formulae*

$$G = \left[\frac{\rho_w R T}{e_s D_v' M_w} + \frac{L\rho_w}{k_a' T}\left(\frac{L M_w}{RT} - 1\right)\right]^{-1} \tag{A1}$$

where the modified diffusivity ($D_v'$) and thermal conductivity ($k_a'$) of water vapor in air account for non-continuum effects
(Seinfeld and Pandis, 2006) and are described as follows

$$D_v' = \frac{D_v}{1 + \frac{D_v}{a_c r}\sqrt{\frac{2\pi M_w}{RT}}} \tag{A2}$$

$$k_a' = \frac{k_a}{1 + \frac{k_a}{a_T r \rho_a c_p}\sqrt{\frac{2\pi M_a}{RT}}} \tag{A3}$$

5  where the thermal accommodation coefficient ($a_T$) is taken as 0.96 (Nenes et al., 2001). Additional sensitivity tests of CDNC
to $a_T$, ranging from 0.1 to 1 (Shaw and Lamb, 1999), were conducted and the resulting droplet concentrations indicate little
sensitivity to this input parameter (not shown here).

The hygroscopicity parameter ($\kappa$) is adopted to characterize aerosol chemical composition on CCN activity according to $\kappa$-
Köhler theory (Petters and Kreidenweis, 2007). $S_{eq}$ for droplets in the $i^{th}$ bin ($i = 1, 2,..., nbin$) can be written as

$$S_{eq} = \frac{r_i^3 - r_{c,i}^3}{r_i^3 - r_{c,i}^3(1 - \kappa_i)}\,exp\left(\frac{2 M_w \sigma_w}{R T \rho_w r_i}\right) - 1 \tag{A4}$$

10  where $r_{c,i}$ and $r_i$ are the radius of the dry aerosol particle and the corresponding growing droplet, respectively. Droplet surface
tension ($\sigma_w$) is a function of the parcel temperature (Pruppacher and Klett, 1997).

$$\alpha = \frac{g M_w L}{c_p R T^2} - \frac{g M_a}{RT} \tag{A5}$$



$$\gamma = \frac{pM_a}{e_s M_w} + \frac{M_w L^2}{c_p R T^2} \tag{A6}$$

Liquid water content (g m$^{-3}$):

$$LWC = \frac{4\pi}{3} \rho_w \sum_{i=1}^{bins} N_i r_i^3 \tag{A7}$$

Quasi-steady approximation of supersaturation $S_{qs}$ (Pinsky et al., 2013):

$$S_{qs} \approx \frac{A_1 V}{4\pi D_v N \bar{r}} \tag{A8}$$

where $\bar{r}$ is the average droplet radius, and $N$ is the total droplet number concentration.

$$A_1 = \frac{g}{R_a T} \left( \frac{L R_a}{c_p R_v T} - 1 \right) \tag{A9}$$

**Appendix B**

**1. Sensitivity to environmental conditions**

To account for the uncertainties associated with the environmental condition from WRF and examine its impact on cloud formation, one additional simulations was conducted with modified temperature profiles at the lowest 2 km above CBH (1,270 m), as displayed in Fig. B1. Here, we adjusted the original lapse rate (-4.1 °C km$^{-1}$ from the WRF sounding, Fig. 10b) to -7 °C km$^{-1}$ ($\Gamma_1$) for 1,270–2,200 m. The lapse rate for 2,200–3,200 m was changed to -4 °C km$^{-1}$ to keep the ambient temperature below CBH and above 3,200 m unchanged. As expected, deeper clouds are formed in the modified environment, representing conditionally unstable atmosphere. As expected, LWC is significantly enhanced and droplet growth is faster under fast cooling condition in the atmosphere.

**2. Sensitivity to initial updraft velocity**

Cloud dynamics also play a crucial role in the microphysical evolution of cumulus clouds. One major parameter in the cloud dynamical field is the updraft velocity. In accordance with the observed vertical velocities by the aircraft and the W-band radar (see Fig. S12b), a reasonable variability in the initial updraft velocity at cloud base is introduced to assess its effects on the parcel supersaturation and cloud droplet concentrations, as shown in Fig. B2. By varying the initial updraft in a range of 0.1–1.5 m s$^{-1}$, simulated results display similar vertical velocities at the observation levels, which are still higher than the measured range (not shown here). As expected, slight increases in maximum supersaturation are resulted from larger initial updraft velocities, thus leading to slight enhancement of total droplet numbers. The simulated spectra show a slightly shift towards larger drop sizes due to weaker updrafts, which allow more time for cloud droplets to grow in a rising parcel.



**Data availability:** The IPHEx data are accessible at Global Hydrology Resource Center (GHRC) Distributed Active Archive Center (https://ghrc.nsstc.nasa.gov/home/field-campaigns/iphex).

**Competing interests:** The authors declare that they have no conflict of interest.

**Acknowledgements**:   This manuscript is revised after work originally submitted to ACP and published in ACPD as Duan et al. (2017). The work was supported in part by NASA grant NNX16AL16G with Barros and NSF Rapid Response Research (RAPID) Collaborative IPHEx grant with Barros (1442039) and Petters (1442056). Yajuan Duan developed the Duke Cloud Parcel Model (DCPM) and conducted the modeling study under the guidance of Ana Barros. Markus Petters was the lead researcher operating the SMPS/CCN system during IPHEx, and provided level-2 datasets of SMPS/CCN measurements (Sect. 3.1). Barros and Duan wrote the manuscript, and Petters provided comments. The authors thank the UND Citation flight scientists, in particular Michael Poellot, Andrew Heymsfield, and David J. Delene for the flight data and advice with airborne data analysis, Si-Chee Tsay and Adrian Loftus for the deployment and operation of the ACHIEVE instruments and W-band radar calibrated data, Anna M. Wilson for the deployment and maintenance of Duke's H2F (Haze to Fog) mobility facility (including the PCASP and raingauges) and corresponding data collection and analysis, and Andrew Grieshop for loaning the X-Ray neutralizer for the duration of the study. We also thank Kyle Dawson and John Hader for operating the SMPS/CCN system in the field and Kyle Dawson for help with the processing of SMPS/CCN data set (Sect. 3.1). We also knowledge computing resources from Yellowstone (*ark:/85065/d7wd3xhc*) at NCAR (allocated to the first author) used for the WRF simulations. The authors are especially grateful to Neil Carpenter from the Maggie Valley Sanitary District for his support of IPHEx activities.

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



**Table 1: Cloud parcel models with detailed microphysics from the literature and in this study (Duke CPM). NA denotes information is not described in the reference paper.**

| Parcel model | Binning | Condensation | Coalescence | Entrainment | Numerics |
|---|---|---|---|---|---|
| Abdul-Razzak et al. (1998) | Discrete | Leaitch et al. (1986) | Not included | No included | LSODE solver (Hindmarsh, 1983) |
| Cooper et al. (1997) | Moving discrete | Fukuta and Walter (1970) | Modified Kovetz and Olund (1969) | Not included | Fifth-order Runge-Kutta (adaptive-size) |
| Flossmann et al. (1985) | Discrete | Pruppacher and Klett (1978) | Berry and Reinhardt (1974) | Lateral homogeneous bubble model | NA |
| Jacobson and Turco (1995) | Hybrid discrete | Jacobson and Turco (1995) | Jacobson et al. (1994) | Not included | SMVGEAR (Jacobson and Turco, 1994) |
| Kerkweg et al., (2003) | Discrete | Pruppacher and Klett (1997) | Bott (2000) | Lateral homogeneous bubble model | NA |
| Nenes et al. (2001; 2002) | Moving discrete | Pruppacher and Klett (1997); Seinfeld and Pandis (1998) | Not included | Not included | LSODE solver (Hindmarsh, 1983) |
| Pinsky and Khain (2002) | Moving discrete | Pruppacher and Klett (1997) | Bott (1998); turbulent effect on drop collision | Not included | NA |
| Snider et al. (2003) | Discrete | Zou and Fukuta (1999) | Not included | Not included | NA |
| *Duke CPM* | Moving discrete | Pruppacher and Klett (1997); Seinfeld and Pandis (2006) | Bott (1998); turbulent effect on drop collision | Lateral homogeneous bubble/jet model | Fifth-order Runge-Kutta (adaptive-size) |



**Table 2. Lognormal fit parameters characterizing the aerosol number distribution of four modes. Note N = total number of aerosol particles per cm³; Dg = geometric mean diameter (µm); σg = geometric standard deviation for each mode. $N_{surf}$ and $N_{CBH}$ represent total aerosol number concentrations at the surface and cloud base height (CBH: 1,270 m), respectively.**

| Mode # | $N_{surf}$ (cm$^{-3}$) | $N_{CBH}$ (cm$^{-3}$) | $D_g$ (µm) | $\sigma_g$ |
|--------|------------------------|-----------------------|------------|------------|
| 1 | 1401.9 | 393.7 | 0.076 | 1.63 |
| 2 | 415.7 | 116.8 | 0.195 | 1.35 |
| 3 | 0.300 | 0.084 | 0.750 | 1.30 |
| 4 | 0.300 | 0.084 | 2.200 | 1.40 |



**Table 3. Evaluation of the predicted CDNC from simulations using various condensation coefficients against the averaged observation from the CDP.**

| Condensation coefficient | Prediction[a] (cm$^{-3}$) (1,500 – 1,600 m) | Difference[b] (%) (1,500 – 1,600 m) | Prediction[a] (cm$^{-3}$) (1,600 – 1,750 m) | Difference[b] (%) (1,600 – 1,750 m) |
|---|---|---|---|---|
| 0.002 | 402.7 | 15.3 | 365.9 | -7.90 |
| 0.005 | 385.8 | 10.4 | 350.5 | -11.8 |
| 0.010 | 354.0 | 1.30 | 321.6 | -19.0 |
| 0.015 | 328.5 | -6.00 | 298.5 | -24.9 |
| 0.030 | 281.0 | -19.6 | 255.3 | -35.7 |
| 0.060 | 242.1 | -30.7 | 219.9 | -44.6 |

[a]The averaged CDNC in the predictions for the indicated altitudes.

[b]Difference (%) = 100×(Prediction - Observation)/Observation. Note observation between 1,500 m and 1,600 m AGL (349.4 cm$^{-3}$) is calculated by averaging the five CDNC measurements and observation between 1,600 m and 1,750 m AGL (397.5 cm$^{-3}$) is calculated by averaging the five CDNC measurements.



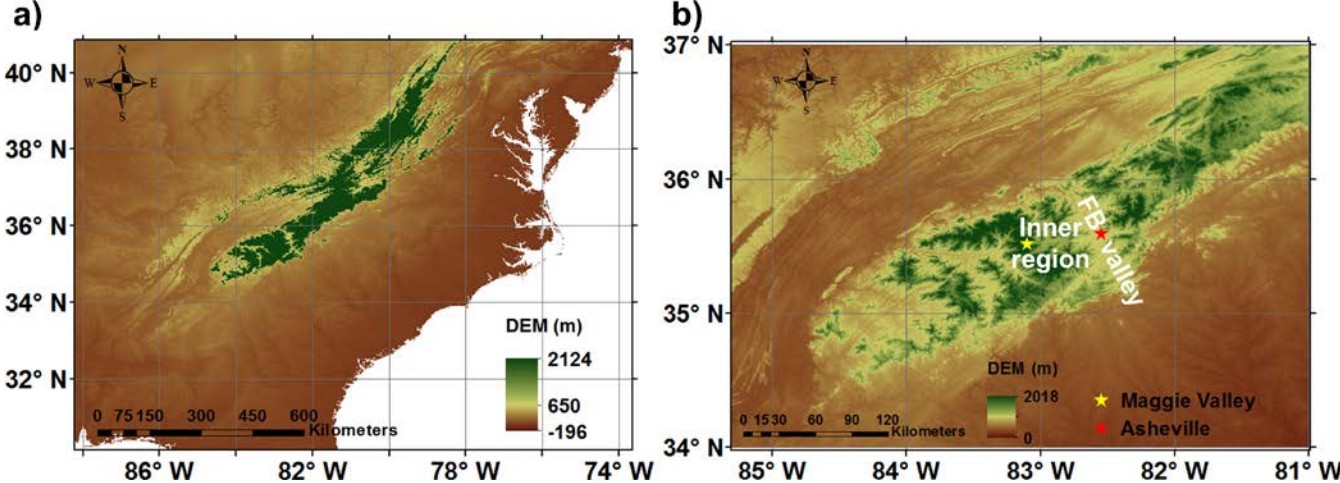

**Figure 1: a) Study region of the IPHEx campaign in the SAM (highlighted in the black box), as shown in context of a large scale map of the southeastern United States. (b) Topographic map of the SAM including the two ground-based IPHEx observation sites referred to in this study. FB valley denotes French Broad valley.**



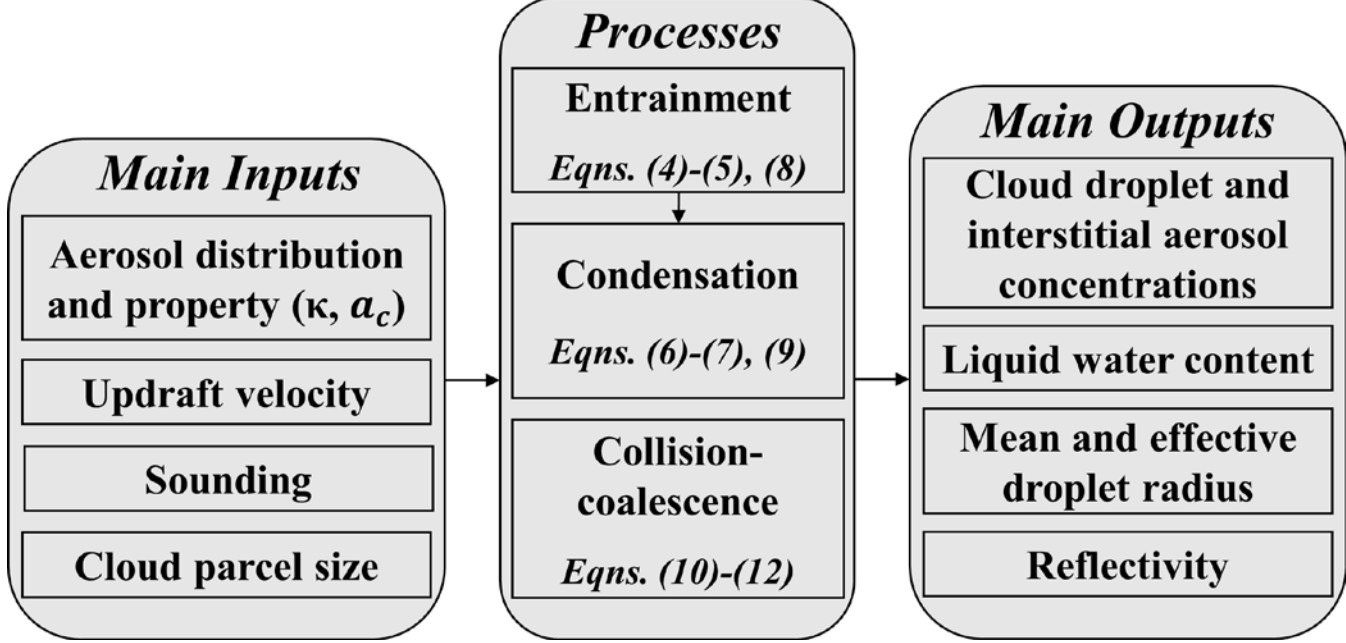

**Figure 2: Flowchart of the main inputs, microphysical processes, and main outputs of the DCPM. Equation numbers refer to formulae in Sect. 2.**



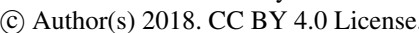


**Figure 3:** a) **Lowest cloud transect of the UND Citation flight track on 12 June 2014. The in-cloud observations are identified as white plus signs, and MV is marked by the black asterisk. For left to right in the map, ECR denotes Eastern Cherokee reservation, MP denotes Mount Pisgah, and FB denotes French Board valley. b) Updraft velocity variations of the targeted in-cloud region, denoted by IC in (a). The in-cloud samples were collected at 1-Hz (~ 90 m in flight distance) resolution. Cloud droplet concentrations of the in-cloud samples in IC (b) with low (0–1 m s$^{-1}$) and high (1–2 m s$^{-1}$) updrafts are shown in (c) and (d), respectively. The updraft velocity of each sample is indicated in the legend. Dotted lines represent the droplet spectra in the reference sub-region within IC, within yellow shade region in (b).**



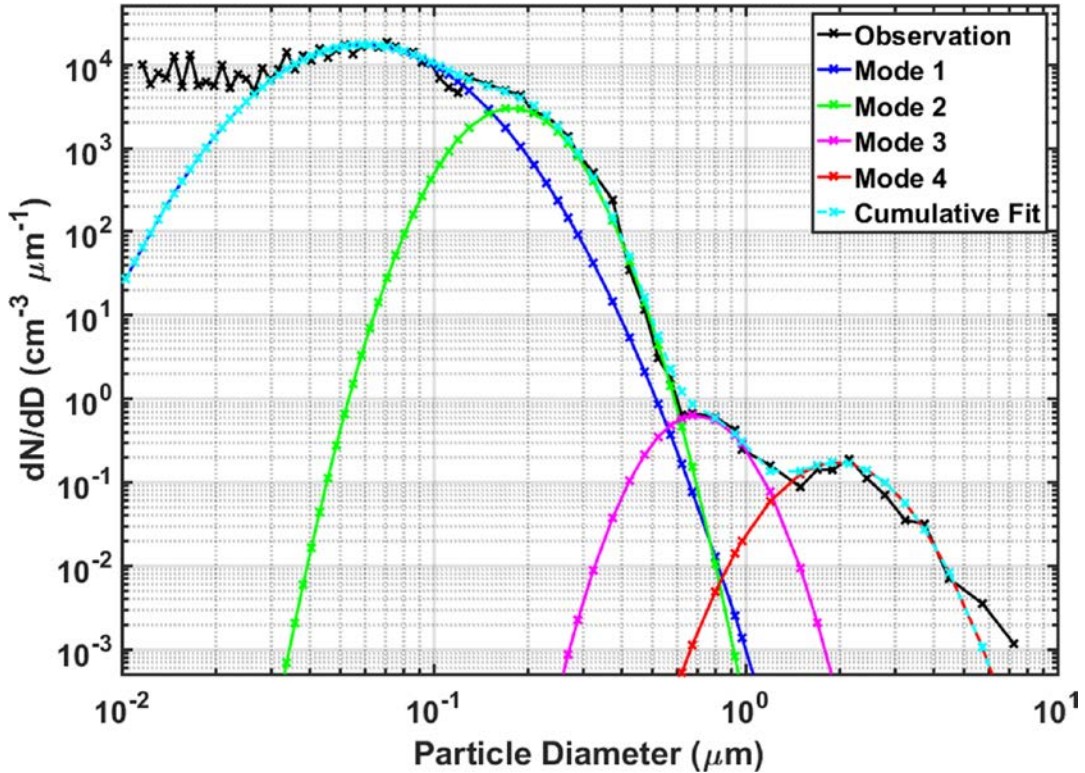

**Figure 4: Mean surface aerosol size distribution fitted by four lognormal functions. Observations are merged from the SMPS and PCASP, and are averaged during the first 10 mins (12:14 LT – 12:24 LT) of the 12 June flight. Fitted parameters (total number concentration, geometric mean diameter, and geometric standard deviation) for each mode are summarized in Table 2.**



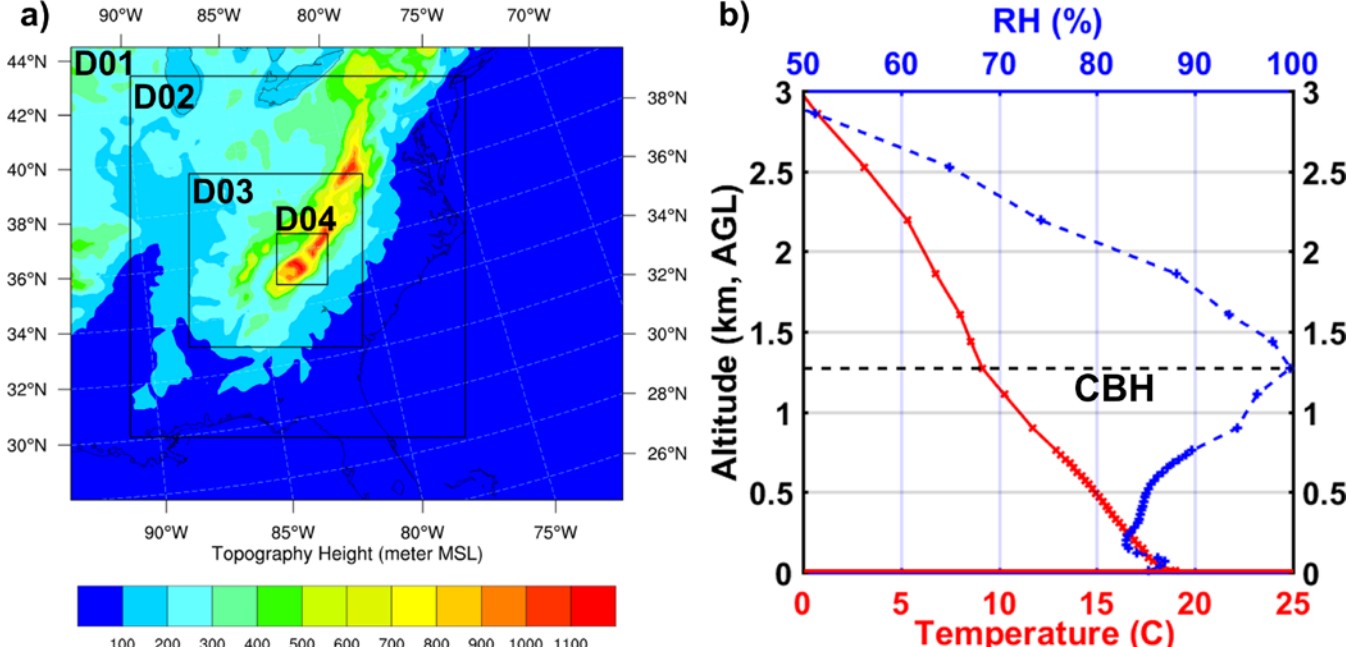

**Figure 5: a) WRF model configuration of four one-way nested domains at 15-, 5-, 1.25-, 0.25-km grid resolution, respectively. b) Vertical profile of temperature (red solid line) and relative humidity (dashed blue line) from the spatially-averaged WRF sounding columns at IC (see its location in Fig. 3a). The horizontal dashed line depicts CBH = 1,270 m AGL.**



**Figure 6: Sensitivity of the updraft velocity (a), supersaturation (b), total drop concentration (c), and LWC (d) to the variations in the condensation coefficient ($a_c$) as compared to the airborne observations (marked by black crosses). The horizontal dashed line depicts CBH. In (b), the quasi-steady approximation of supersaturation is calculated based on observed temperature. It should be kept in mind that airborne measurements of temperature in clouds are subject to large uncertainties, thus rendering the derivation of supersaturation unreliable.**



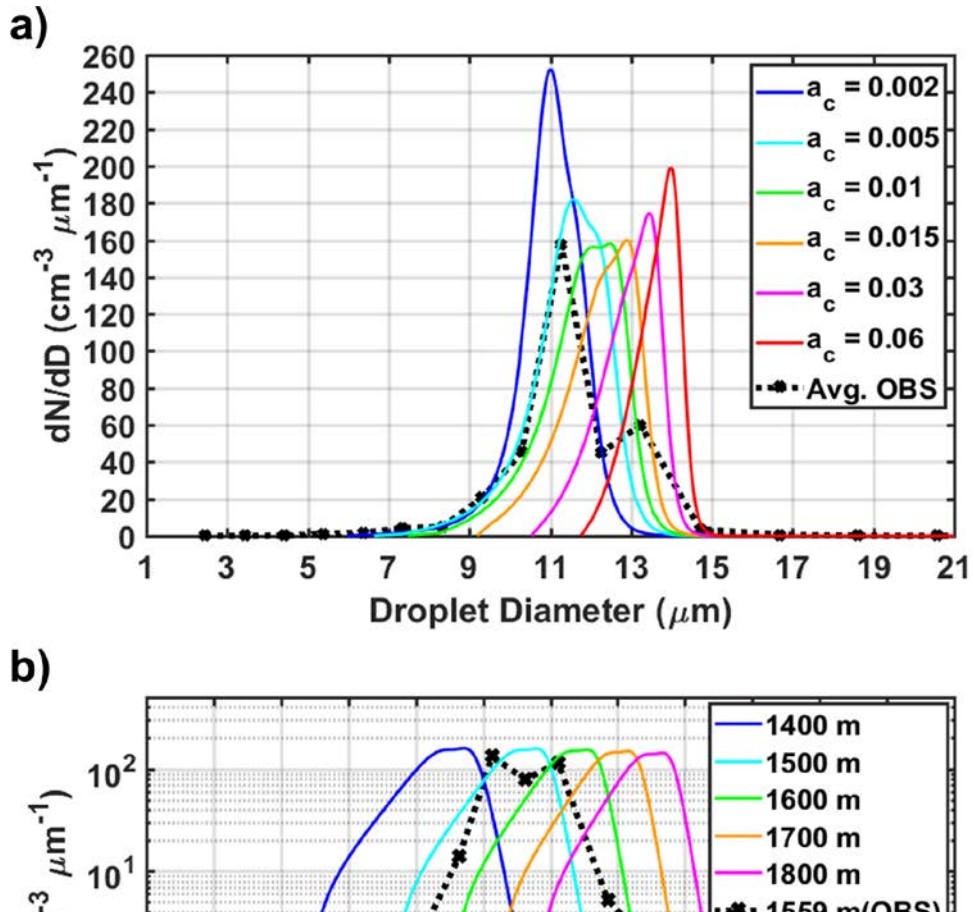

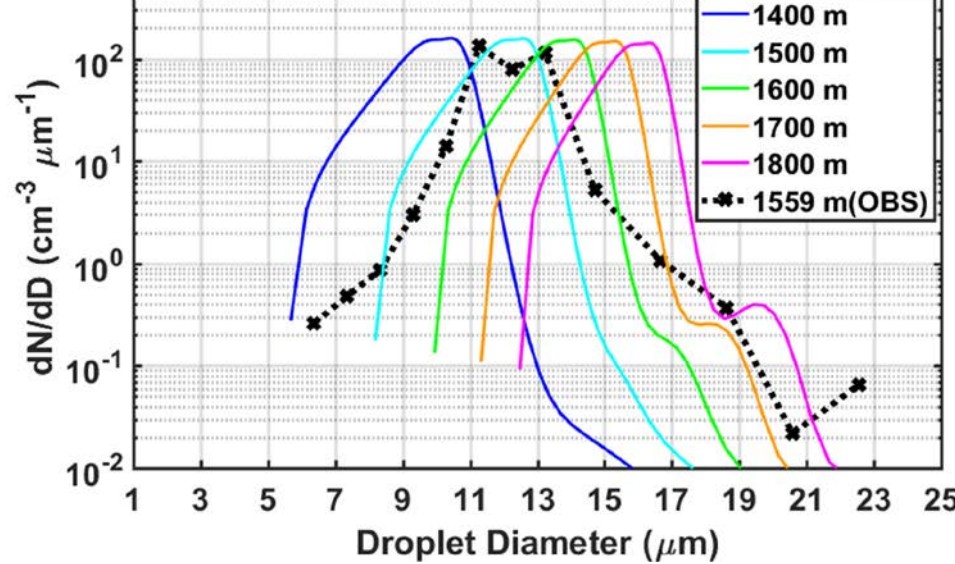

**Figure 7: a) Sensitivity of simulated droplet spectra at 1,500 m (solid lines) to the variations in $a_c$. The black dotted line reflects the average of five droplet spectra observed by the CDP (dotted lines in Figs. 3c and d) between 1,500 m and 1,600 m AGL. b) Simulated evolution of cloud droplet spectra at 1,400 m, 1,500 m, 1,600 m, 1,700 m, and 1,800 m altitude assuming $a_c$= 0.01. The black dotted line denotes the observed droplet spectrum at 1,559 m that has similar total CDNC and LWC as the simulation with $a_c$ = 0.01 at the same altitude.**



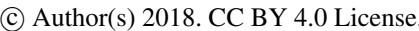

**Figure 8: Sensitivity of the total drop concentration (a) and LWC (b) to the variations in the initial parcel radius (R) considering lateral entrainment as a bubble model and a jet model. In (a) and (b), the airborne observations are marked by black crosses, and the horizontal dashed line depicts CBH. c) Predicted droplet spectra at three altitudinal levels (1,500 m: solid line, 1,600 m: dotted line, and 1,700 m: dashed line) using two parameterization schemes for lateral entrainment: the bubble model with R = 500 m (base case, grey lines), R = 300 m (cyan lines), and R = 1,000 m (green lines); the jet model with R = 500 m (red lines). The black dotted line reflects the average of five droplet spectra observed by the CDP (dotted lines in Figs. 3c and d) between 1,500 m and 1,600 m AGL.**



**Figure 9: Sensitivity of the total drop concentration (a), LWC (b), and droplet spectra (c) at three altitudinal levels (1,500 m: solid line, 1,600 m: dotted line, and 1,700 m: dashed line) to the variations in initial aerosol concentrations at cloud base, as represented by different values of the scale height (H$_S$). In (a) and (b), the airborne observations are marked by black crosses, and the horizontal dashed line depicts CBH. The black dotted line in (c) reflects the average of five droplet spectra observed by the CDP (dotted lines in Figs. 6c and d) between 1,500 m and 1,600 m AGL.**





**Figure 10:** Sensitivity of the total drop concentration (a), LWC (b), and droplet spectra (c) at three altitudinal levels (1,500 m: solid line, 1,600 m: dotted line, and 1,700 m: dashed line) to variations in hygroscopicity parameter (κ). In (a) and (b), the airborne observations are marked by black crosses, and the horizontal dashed line depicts CBH. The black dotted line in (c) reflects the average of five droplet spectra observed by the CDP (dotted lines in Figs. 3c and d) between 1,500 m and 1,600 m AGL.



**Figure B1: Vertical profiles of the supersaturation (a) and LWC (b) for simulations with the original WRF sounding (grey lines) and modified ambient temperature (blue lines). In (b), the airborne observations are marked by black crosses, and the horizontal dashed line depicts CBH. c) Predicted droplet spectra at three altitudinal levels (1,500 m: solid line, 1,600 m: dotted line, and 1,700 m: dashed line) to the variations in the environmental conditions. The black dotted line reflects the average of five droplet spectra observed by the CDP (dotted lines in Figs. 3c and d) between 1,500 m and 1,600 m AGL.**



**Figure B2: Sensitivity of the supersaturation (a), total drop concentration (b), and droplet spectra (c) at three altitudinal levels (1,500 m: solid line, 1,600 m: dotted line, and 1,700 m: dashed line) to the variations in the initial updraft velocity ($V_0$) at cloud base. In (b), the airborne observations are marked by black crosses, and the horizontal dashed line depicts CBH. The black dotted line in (c) reflects the average of five droplet spectra observed by the CDP (dotted lines in Figs. 3c and d) between 1,500 m and 1,600 m AGL.**





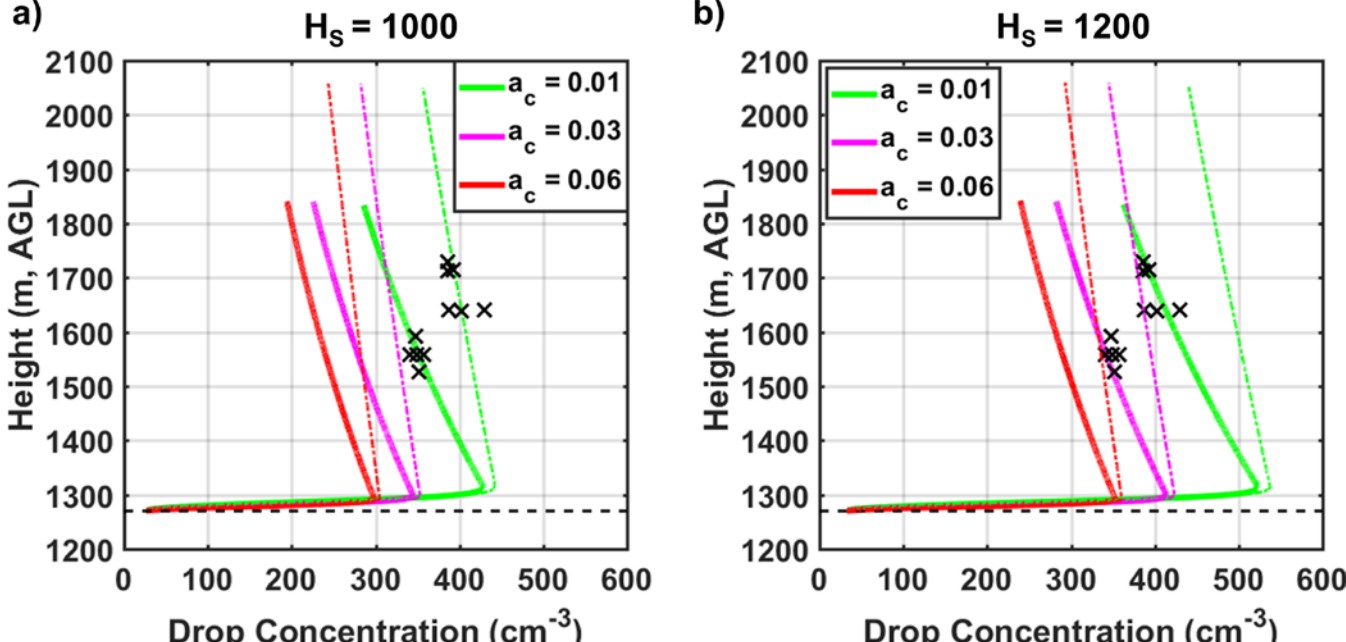

**Figure B3: Sensitivity of the total cloud drop concentration to the variations in condensation coefficient and entrainment strength (strong: R = 500 m, solid thick lines; weak: R = 1,500 m, dash-dot thin lines) assuming different initial aerosol concentrations at cloud base (a: $H_S$ = 1,000 m; b: $H_S$ = 1,200 m). The airborne observations are marked by black crosses, and the horizontal dashed line depicts CBH.**