# Peer review of "Understanding aerosol-cloud interactions through modelling the development of orographic cumulus congestus during IPHEx"

_Atmospheric Chemistry and Physics, 2018_

## Author Comment (AC1) · 19 May 2018

The present manuscript is a revised version of Duan et al. (2017), which was submitted previously to ACPD. That manuscript was reviewed by one additional referee, which was not made public at the time. Responses to this second round of reviews are provided below. The posted manuscript here was revised to address the points raised by that referee in addition to the first two referres.

Duan, Y., Petters, M. D., and Barros, A. P.,2017. Understanding aerosol-cloud inter-actions in the development of orographic cumulus congestus during IPHEx, Atmos. Chem. Phys. Discuss., DoI:10.5194/acp-2017-396.

[Figure]

Please also note the supplement to this comment:
https://www.atmos-chem-phys-discuss.net/acp-2018-419/acp-2018-419-AC1-
supplement.pdf

———————————————————

[Figure]

**Supplement:**

Reply to Reviewer #3

The Referee's comments were taken into consideration fully in the revised manuscript. Whereas we disagree with the overall assessment regarding the manuscript contributions (the cloud parcel model presented is the only publicly available parcel model that includes entrainment and can produce the vertical distribution of CDNC in a consistent numerical framework to addresses the multiscale time-space challenge; the sensitivity results presented for the competitive interference of entrainment strength and condensation coefficient are discussed for the first time; and much of the IPHEx data analysis and related modeling results has not been published previously), we do appreciate the points raised regarding the need to highlight the science goal and improve manuscript organization, and we made a conscientious effort to address all issues raised. Thank you.

Item-by-item replies are in blue. The Referee's comments are in black.

I think the authors did a lot of work for the study. They developed a new cloud parcel model, analyzed observational data, and did parcel model simulations to look at the sensitivity of cloud properties to aerosol and thermodynamic parameters. However, the current version of the paper does not have clear science goal and is not organized as a coherent research, and the results do not present much new understandings. These major comments are reflected by the specific comments below. I think quite a lot work is needed to have the paper of the scientific significance for ACP.

Substantial revisions were made in the Introduction Section to explicitly state the scientific objectives of this study, as well as throughout the manuscript to present a clear and coherent structure of the manuscript. The manuscript title was also slightly changed to reflect the modelling focus as follows:

*"Understanding aerosol-cloud interactions through modelling the development of orographic cumulus congestus during IPHEx"*

The main objective of this manuscript is to investigate the vertical evolution of cloud microphysical structure (e.g., number concentration and droplet spectra) by developing a new cloud parcel model (CPM) that explicitly solves the cloud microphysics of condensation, collision-coalescence, and lateral entrainment processes with temporal and vertical resolution consistent with the range of scales of the governing processes (μm-m; ms-s). The long-term science goal of this new CPM was explained in the revised manuscript (Pg. 4, Lines 1 - 9). The long-term goal for this new CPM is to be coupled to an existing stochastic rainfall column model, describing the dynamic evolution of raindrop microphysics (bounce, collision-coalescence, and breakup mechanisms) between cloud base and the ground surface (Prat and Barros, 2007;Prat et al., 2012). The purpose of this coupled system is to investigate low-level precipitation enhancement induced by seeder-feeder interactions (SFI) among multilayer clouds generally, and between locally initiated or propagating convective clouds and low-level boundary layer clouds in particular. Previous research (Wilson and Barros, 2014) showed that SFI can increase the intensity of rainfall by up to one order of magnitude in the SAM and explain the observed mid-day peak in rainfall. Thus, the ability to predict the evolution of cloud formation and its vertical structure of droplet

size distribution (DSD) in this region is of paramount interest. To maintain a coherent organization of the manuscript, the Section on multi-parcel simulations was removed from the revised manuscript and the discussion on the sensitivity to hygroscopicity was moved from Appendix to Sect. 4.2.4 in the revised manuscript. Replies to specific comments are addressed next.

1. Introduction, first paragraph, does not include discussion of any recent studies on aerosol-cloud interactions for orographic clouds (there are quite a lot studies after 2012 on this topic and significant understandings are gained). It is necessary to discuss the most recent progresses in this area and introduce what the major problem is.

The Referee's point is well taken. More discussion of recent studies from the peer-reviewed literature on aerosol-cloud interactions of orographic clouds were added in Sect. 1 (Pg. 2, Lines 4-16).

2. Page 2, Line 26-28, the statement is not correct.

We have clarified this statement in Section 1 (Pg. 2, Lines 22-25) in the revised manuscript. Although detailed 2-D and 3-D models that explicitly resolve cloud formation and microphysical evolution to varying degrees of completeness have been developed (Fan et al., 2009;Leroy et al., 2009;Muhlbauer et al., 2010), relatively large temporal and spatial resolutions, and coarse spectral resolution of aerosols and cloud droplets cannot explicitly solve the nonlinear stochastic coalescence-breakup dynamics that govern the vertical microstructure of clouds and rainfall. Even when high vertical resolution is used such as in Wilson and Barros (2017), where 100 vertical layers were used with at least 14 in the lowest 1 km. This presents a critical modeling challenge in capturing the variations in vertical velocity profiles of hydrometeors at high vertical resolution (~ 10 m, used in the rainfall column model), thus unable to explain low-level precipitation enhancement induced by SFI in the SAM.

3. Introduction, second paragraph on P.4, I am not clear of the purpose of paper. What is the purpose of replicating aircraft microphysical observations with a cloud parcel model? What is the big picture here in terms of goals? Do you want to gain something for better parameterization for realistic modeling or gain new understandings about important physical factors impacting aerosol-cloud interactions over the complex terrain? The paper needs further work in terms of organization toward either direction. In addition, cloud parcel models are not the tool meant for reproducing observed cloud properties, particularly for the clouds developed over the complex terrain. The dynamics of valley-mountain circulations, which cannot be simulated with parcel models, are important to determine cloud properties and aerosol-cloud interactions. Even if you reproduce some aspects of the observed clouds, it could just because of a wrong reason considering the simplicity of cloud parcel model in both aerosol setup and dynamics.

The Referee's point is well taken. Indeed, the authors recognized the complicated flow dynamics in complex terrain. In the manuscript, high-resolution (250 m horizontal grid size) model simulations were conducted to illustrate the ridge-valley circulations in this region and detailed discussion can be found in the supplementary material Sect. S3. Understanding the complex

dynamics of ridge-valley circulations in this region, cloud segments to perform the CPM study were carefully selected by screening the cloud droplet spectra obtained from the cloud droplet probe (CDP) during IPHEx, as stated in Sect. 3.3 (Pg. 10, Line 27-32). Criteria listed in Conant et al. (2004) were used to eliminate regions influenced by mixing and other unresolved mechanisms and only one in-cloud region (IC) was selected as it satisfies Conant et al.'s requirements.

To heed to the Referee's concerns regarding the limitations of a simple CPM, the purpose of this manuscript is on understanding the vertical development of clouds leading to the mid-day precipitation peak in the inner SAM region, and to investigate the quantitative impact of key aerosol-cloud interactions (ACI) modeling parameters and initial assumptions on early cloud development. Model results were compared to the airborne observations of cloud microphysics as good agreement was achieved between the two, thus it provides more confidence in the findings from the sensitivity tests to determine the controlling factors in the microphysical evolution of clouds at early stages in the SAM.

4. P15, Line 22-24, I do not think Fig. 11a shows that the simulated vertical velocity is consistent with the observations in either profile or magnitude.

As shown in Fig. 6a (Fig. 11a in the previous manuscript), the simulated updraft velocities at the observation levels are consistent with the general trend of airborne measurements, which decrease with height. The figure points out the negligible influence of large $a_c$ values on the simulated profiles of vertical velocity. The purpose of this figure is not to reproduce the observed vertical velocity in the cloud.

5. P15-16, the different droplet concentration between above and below 1.6 km could be because of different aerosol composition (implying different condensation coefficients) and/or concentration in reality. So, what is the purpose of playing different condensation coefficients to have a good fit for below and above that altitude? In reality, what we need to understand if there are dynamics and thermodynamics difference first, then look at if there are differences in aerosol properties.

As stated in the manuscript (Pg. 13, Line 30- Pg. 14, Line 2), based on the drop spectra there are likely two clusters of air parcels at different levels above ground, but there is no clear distinction of observed vertical velocities between these two clusters. Thus, the hypothesis that the differences in droplet concentrations and spectra might be a result of different condensation coefficient values. More importantly, the purpose of varying the $a_c$ values is to investigate the sensitivity of model results to condensation coefficient, not to calibrate the model result by fitting with the observations.

6. Figure 12a shows that the cloud parcel did not get the droplet spectral right at 1.5 km. Obviously, there are two modes in observations, a cloud droplet mode peaking at 11 um and a raindrop model peaking at ~14 um. The second mode indicates rain formation starts at this altitude. This indicates cloud parcel model only can get a single mode and has deficiency in getting complicated drop spectrum (might be due to simple updraft dynamics).

At early stage of cloud development, unimodal spectra were produced due to the dominant role of condensation. As shown in Fig. 7b, the droplet growth from effective collision-coalescence process results in a secondary mode at larger sizes at later stages. Therefore, the CPM is able to produce bimodal droplet spectra as observed in clouds. At the targeted IC region, precipitation-sized drops were not present in concurrent cloud observations (highlighted in dark blue shade in Fig. S14d), thus the second mode is not indicative of rain formation. The lack of bimodal droplet spectra was discussed in Sect. 5. As introduced by Pinsky and Khain (2002), in-cloud nucleation of new droplets from interstitial aerosols can form a secondary mode at a smaller droplet size when the parcel supersaturation higher up in the cloud exceeds the cloud base maximum. However, this is not the case here by using the WRF sounding input.

7. P. 17 line 15-17, the statement "uncertainties of the assumed environmental thermodynamic conditions (e.g., temperature) impose significant constraints in the vertical development of clouds, thus posing as a significant challenge in cloud modeling study" only states the basic concept of convective cloud development. It should not be stated as a major conclusion of the research.

The statement (Pg. 15, Lines 7-9 in revised manuscript) was concluded from the sensitivity results by modifying the environmental conditions from WRF, as discussed in Appendix B1. The statement is not presented as a major conclusion of this study.

8. Section 4.2.2, need a transition about the purpose of looking at the impact of entrainment strength. In addition, the parcel size is only a parameter existing in the parcel model framework. Physically a parcel size is already determined by the environmental conditions and you should not have this freedom for a certain environment. To be useful for realistic 3-D model simulations, we need to know what this parameter means in 3-D model framework. Can it be translated to model resolution?

Each subsection in Sect. 4.2 discusses sensitivity results to a different key parameter in the model. The sensitivity experiments in Sect. 4.2.2 are to examine the role of entrainment strength in modifying the cloud microphysical properties, and they are not used for determining a parcel size under the given environment.

9. P.19, first paragraph, the statement "aerosols in the atmosphere exhibit a significant space-time variability especially in regions of complex terrain with heterogeneous mixing by different ventilation processes in addition to the possibility of remote transport, all of which can contribute to the diverse cloud droplet spectra observed across the cloud transect (see Figs. 8a-c). This cannot be captured by the current model simulations that assume a homogenous aerosol distribution at cloud base". Exactly. It seems to me that research started from this problem and then concluded the same problem. Then what is the use of the study? Currently the results show droplet spectrum is sensitive to aerosol properties and thermodynamic properties. These are well-studied already in the field.

As stated in the reply to Comment #3, the objective of this manuscript is to investigate the vertical development of cloud microphysical properties in complex terrain by implementing a new

entraining CPM. Ultimately the vision is to track the evolution of local (within column) clouds from CCN activation through cloud formation, sustained cloud development, and rainfall. Sensitivity tests were conducted to assess the quantitative impact of key ACI modeling parameters and initial assumptions on cloud microphysical development at early stages. Model results of cloud droplet number concentration (CDNC), liquid water content, and droplet spectra show good agreement with airborne observations from IPHEx. Discrepancies between model results and observations were further discussed with respect to limitations of the CPM and challenges of its application in complex terrain. In particular, this study introduces a new entraining CPM and the model is successfully applied to explore the influence of entrainment strength on the cloud microphysical evolution, which are new contributions to the field.

10. P.19, Line 22-23, "Based on the sensitivity tests, the cloud spectra observed in the inner region of the SAM for early development of cumulus congestus on 12 June are reproduced better by a relatively low value of $ac$ (0.01).", I do not think you can conclude it since your initial aerosol size distribution and composition around cloud base are not constrained by observations. You can easily achieve a similar droplet activation rate for different $ac$ by adjusting initial aerosol size distribution.

The Referee's point regarding the lack of observation constraints due to limited data in our case is well taken. Because of IPHEx's emphasis on precipitation microphysics and specifically the vertical distribution of hydrometeors, aerosol measurements were limited compared to other campaigns when the focus has been strictly on aerosols, in particular near and at cloud base. In this study, there are no measurements at cloud base or near cloud base proper as the focus was on precipitation microphysics during the field campaign. As noted in Sect. S3 (Pg. 4, Line 5 - Pg. 5, Line 2), high-resolution model results show there is significant horizontal wind shear with 3D circulations including valley winds, thermal winds, ridge-valley circulations, and well defined southerly mesoscale mid-level transport patterns that result in complex horizontal and vertical wind structure very different therefore from the classical convective boundary layers in flat terrain. Given the complexity of the 3D circulations, especially the horizontal winds, the regional convergence patterns and the potential role of advective venting, and the nearly self-similar regional distribution of vertical velocities at the mesoscale, initial aerosol concentration at cloud base is extrapolated vertically from the surface aerosol number concentrations at MV by assuming an exponential decay with a scale height ($H_S$). This assumption is in keeping with previous studies using mesoscale models (Iguchi et al., 2008;Muhlbauer and Lohmann, 2008) and cloud parcel models (e.g., Eichel et al., 1996) and consistent with differences in total aerosol numbers between the two horizontal flight legs at different heights over IC (not shown here). The reference scale height ($H_S = 1000$ m) in this study is within the typical range ($H_S$: 550–1,100 m) for remote continental aerosol type (Jaenicke, 1993), which is a reasonable assumption.

As discussed in Sect. 1 (Pg. 3, Line 16 - 32) and Sect. 4.2.5 (Pg. 17, Line 30 - Pg. 18, Line 4), a relatively low value of $a_c$ (0.01) is consistent with previous findings from laboratory experiments with background air flows, thus not adiabatic conditions. Recognizing that limited measurements were available at cloud base, the conclusion (Pg. 17, Lines 30-32) regarding $a_c$ has been revised

to read "Based on the sensitivity tests, model simulations using a relatively low value of $a_c$ (0.01) exhibit CDNC and spectra consistent with the cloud spectra observed in the inner region of the SAM for early development of cumulus congestus on 12 June." This further highlights the need to have a constraining set of observational inputs in order to validate our findings over the SAM. Additional simulations were conducted assuming well-mixed conditions, and the results are included in the Supplementary Material Sect. S4. Thus, surface aerosol concentrations were used as model input, assuming a well-mixed well-developed convective boundary layer. The results are presented and discussed in Sect. S4 (see the Supplementary Material). Although agreement in total CDNC can be obtained between simulations with larger $a_c$ values and the airborne observations, large discrepancies exist between the predicted and observed activated spectra with implications for precipitation processes. Detailed discussion can be found in Sect. 4.2.3 (Pg. 16, Lines 17 - 29).

11. Section 4.3, I do not get the point of multi-parcel simulation discussion based on the text here. The text only describes the techniques and what the first, second, and third parcel look like. It is not connected with any points presented before.

The Referee's point is well taken. The Section of multi-parcel simulation was removed from the revised manuscript.

12. Section 5, since the point "simulated CDNC also exhibits high sensitivity to variations in initial aerosol concentration at cloud base, but weak sensitivity to aerosol hygroscopicity" is a key conclusion shown in the abstract, the results for weak sensitivity to aerosol hygroscopicity should be presented in the main figure instead of putting in the Appendix. This indicates the results are not well organized. Another evidence to support this point is the authors used 6 figures (Figures 3-8) to just present observational data that they have from the field campaign, while do not make any important points to the paper. The results of different sensitivity tests need to be organized in a coherent and logic way (right now it seems that a bunch of tests are put together and the results are discussed for each set of the tests).

The Referee's point is well taken. The discussion of sensitivity results to aerosol hygroscopicity was moved to Section 4.2.4 in the revised manuscript. Figures 3-5 and 7-8 in the previous manuscript were discussed in the supplementary section (see Sect. S2) and are now Figures S7-S8, S11, and S14-S15 in the revised manuscript.

Thank you

**References**
Conant, W. C., VanReken, T. M., Rissman, T. A., Varutbangkul, V., Jonsson, H. H., Nenes, A., Jimenez, J. L., Delia, A. E., Bahreini, R., Roberts, G. C., Flagan, R. C., and Seinfeld, J. H.: Aerosol-cloud drop concentration closure in warm cumulus, Journal of Geophysical Research: Atmospheres, 109, 10.1029/2003jd004324, 2004.

Eichel, C., Krämer, M., Schütz, L., and Wurzler, S.: The water-soluble fraction of atmospheric aerosol particles and its influence on cloud microphysics, Journal of Geophysical Research: Atmospheres, 101, 29499-29510, 10.1029/96jd02245, 1996.

Fan, J., Yuan, T., Comstock, J. M., Ghan, S., Khain, A., Leung, L. R., Li, Z., Martins, V. J., and Ovchinnikov, M.: Dominant role by vertical wind shear in regulating aerosol effects on deep convective clouds, Journal of Geophysical Research, 114, 10.1029/2009jd012352, 2009.

Iguchi, T., Nakajima, T., Khain, A. P., Saito, K., Takemura, T., and Suzuki, K.: Modeling the influence of aerosols on cloud microphysical properties in the east Asia region using a mesoscale model coupled with a bin-based cloud microphysics scheme, Journal of Geophysical Research, 113, 10.1029/2007jd009774, 2008.

Jaenicke, R.: Tropospheric aerosols, in: Aerosol-cloud-climate interactions, edited by: Hobbs, P. V., Academic Press, 1-31, 1993.

Leroy, D., Wobrock, W., and Flossmann, A. I.: The role of boundary layer aerosol particles for the development of deep convective clouds: A high-resolution 3D model with detailed (bin) microphysics applied to CRYSTAL-FACE, Atmospheric Research, 91, 62-78, 10.1016/j.atmosres.2008.06.001, 2009.

Muhlbauer, A., and Lohmann, U.: Sensitivity studies of the role of aerosols in warm-phase orographic precipitation in different dynamical flow regimes, Journal of the Atmospheric Sciences, 65, 2522-2542, 2008.

Muhlbauer, A., Hashino, T., Xue, L., Teller, A., Lohmann, U., Rasmussen, R. M., Geresdi, I., and Pan, Z.: Intercomparison of aerosol-cloud-precipitation interactions in stratiform orographic mixed-phase clouds, Atmospheric Chemistry and Physics, 10, 8173-8196, 10.5194/acp-10-8173-2010, 2010.

Pinsky, M. B., and Khain, A. P.: Effects of in-cloud nucleation and turbulence on droplet spectrum formation in cumulus clouds, Quarterly Journal of the Royal Meteorological Society, 128, 501-534, 2002.

Prat, O. P., and Barros, A. P.: A Robust Numerical Solution of the Stochastic Collection–Breakup Equation for Warm Rain, Journal of Applied Meteorology and Climatology, 46, 1480-1497, 10.1175/jam2544.1, 2007.

Prat, O. P., Barros, A. P., and Testik, F. Y.: On the Influence of Raindrop Collision Outcomes on Equilibrium Drop Size Distributions, Journal of the Atmospheric Sciences, 69, 1534-1546, 10.1175/jas-d-11-0192.1, 2012.

Wilson, A. M., and Barros, A. P.: An Investigation of Warm Rainfall Microphysics in the Southern Appalachians: Orographic Enhancement via Low-Level Seeder–Feeder Interactions, Journal of the Atmospheric Sciences, 71, 1783-1805, 10.1175/jas-d-13-0228.1, 2014.

Wilson, A. M., and Barros, A. P.: Orographic Land-Atmosphere Interactions and the Diurnal Cycle of Low Level Clouds and Fog, Journal of Hydrometeorology, 10.1175/jhm-d-16-0186.1, 2017.

---

## Referee Comment (RC1) · Anonymous Referee #1 · 18 Jun 2018

This study uses a warm-phase cloud parcel model that simulates the cloud droplet activation by aerosol particles, water condensation, collision-coalescence, and lateral entrainment processes to investigate aerosol-cloud interactions in one of the IPHEx cases. The comparisons between the in-situ observations and parcel model sensitivity results indicate that the condensation coefficient is the most important parameter determining cloud droplet number concentrations, liquid water content and size distribution. The cloud development is also sensitive to entrainment and aerosol concentration at cloud base but is not sensitive to aerosol hygroscopicity.

The manuscript is not very well organized. Readers have to resort to different locations

(main contexts, appendix and supplemental materials) for important and necessary information. The development of a cloud parcel model is a non-trivial work. However, such a tool should be used to answer critical scientific questions. There are few new scientific points being discovered in this work, which does not makes it qualified for the ACP publication. I would suggest the authors to submit this manuscript to other journals that have an emphasis on model development or test.

I listed some of my concerns in the following: 1. If the condensation coefficient is the dominant parameter for cloud development and evolution in the early stage, how do the authors choose a value or develop a parameterization to provide a reasonable value for the more detailed 3D simulations? 2. As the authors pointed out, lateral entrainment is not appropriate especially for orography influenced clouds. A more appropriate entrainment scheme might be needed for the work. 3. The equation 8 is not clear to me. Why the droplet number in the ith bin is determined by the aerosol number concentration in the ith bin in entrainment? 4. Page 10, lines 14-15, why the corrected CDP spectra that is shifted to smaller size provided confidence in the performance of the CDP probe during the IPHEx campaign? 5. Page 11, line 31, the volume ratio of 1.026 is not correct. It should be 1.021. 6. Figure 7b looks wrong to me. The condensation process is inversely related to the droplet size. But in Fig. 7b, the entire DSD shifts to the right without showing any narrowing of the DSD. 7. Since many parameters impact the cloud development, there are multiple combinations of these parameters to provide the same cloud development trajectory. How can the authors justify which combination is the right one?

---

## Referee Comment (RC2) · Anonymous Referee #2 · 20 Jun 2018

In this study, a new advanced cloud parcel model, which includes description of condensation, collision-coalescence, and lateral entrainment processes, is developed. The model is designed so as to be coupled with the rainfall microphysics column model describing the evolution of raindrops size distributions. In wide aspect the model will be used for investigation of aerosol-cloud interaction. In order to set initial and boundary meteorological and aerosol conditions for model study authors also analyzed the results of measurements, obtained in the IPHEx campaign in this paper. The results of the supplemental simulations by the WRF model are also applied to specify sounding, used in the parcel model. The data obtained in the IPHEx campaign also serves as a reference in the comparison of model and measured results.

[Figure]

The main output of the study illustrated in the figures 6-10 and B1-B3, is the results of sensitivity simulations, that show the variability of the different microphysical and thermodynamic quantities such as droplet concentration profiles, LWC profiles, supersaturation profiles, droplet size distribution (DSD) to variation of condensation coefficient, to type of parcel model (bubble vs jet), to initial aerosol concentration and hygroscopicity parameter as well as to environment conditions and initial updraft velocity. Comparison with experimental data allowed to choose optimal parameters providing minimal discrepancy between model and experimental data. Agreement between modeled and measured DSDs can be characterized as reasonably good.

Several corrections which fall between "major" and "minor" should be done before the manuscript is published in ACP:

1. I think the purpose of this study must be specified more clearly. If this study is a part of a more general study an introduction to the general purpose should be done.

2. Three microphysical processes (condensation, collision-coalescence, and lateral entrainment) are described very accurately in the model. At the same time such processes as droplet sedimentation and ventilation are completely absent from the model. Authors should justify the omission of these processes.

3. Although the model parameters (time steps, number of boxes, etc.) are given throughout the text, it would be desirable to present a table, containing the main model parameters.

4. There is a large number of references to the supplementary material as well as discussion of figures and sections of this material in the paper. These materials are not available for readers, so these parts of the text are not clear. Authors should find a more understandable form for the presentation of these parts of the text.

Remarks:

Page 4, line 12: Incomplete sentence.

Page 6, line 5 : Do I understand correctly that is equal or to or to ? If it is so to write this in the text more clearly.

Page 6, Eq. (6): To note that depends on the index .

Page 6, Eq. (7): I was not familiar with the supersaturation equation written in this form. Provide reference to the derivation of this equation please.

Page 7, line 14: Terminal velocity of droplets (Eq. (12)) and updraft velocity of parcel (Eqs. (4) and (5)) are denoted by the same symbol. To use different symbols.

Page 16, line 3: To replace Fig 13a by Fig 8a.

Page 18, line 16: To replace "wam" by "warm"

Page 20, line 29-30: I think the mode of the largest droplets seen in the Fig. 7 relates with collision process.

---

## Referee Comment (RC3) · Anonymous Referee #3 · 6 Jul 2018

This study presents a new entraining cloud parcel model that includes activation, condensation, collision-coalescence, and lateral entrainment processes. This model was applied to a case study to investigate dominant factors in determining the microphysical development of clouds over complex terrain such as aerosol-cloud interaction during IPHEx campaign. The model was tested for a mid-day cumulus congestus case where aircraft measurements were available. Also, the authors used the measurements from IPHEx campaign and WRF modeling to provide initial conditions of some variables to the cloud parcel model. The authors stated that the modeling results for the reference simulation achieved a good agreement with the cloud droplet number concentration, liquid water content, and droplet size spectra observation few meters above cloud base.

[Figure]

Based on in-situ measurement and model sensitivity results, the authors found that condensation coefficients are a key parameter for this case. In addition, the model is sensitive to entrainment and aerosol concentration at the cloud base. Although the authors performed a good job in the development and analysis of the new model, the manuscript did not present a clear focus on answering new scientific questions related to the aerosol-cloud interactions and in some parts of the manuscript the goal of this work can be misinterpreted as a model development, which makes the current status of this manuscript not scientific significant for ACP publication. As such, I would recommend major revisions.

Specific comments: 1) The manuscript is not well organized, in some parts of the manuscript the readers have to go back and forth between main text and supplementary material, which make the readers confused on whether the information is important or just a supplementary information, one example is the section 3.3 lines 24-31. I would suggest the authors find a more organized way to present those discussions and a better transition between the text, the main figures and the supplementary figures. 2) Page 15 lines 18-19 the authors stated that one of the explanations for the absence of small drops is the uncertainties coming from WRF simulated radiosondes. I can see a possibility that the WRF simulation did not represent correctly the vertical structure of the atmosphere (especially in the lower levels) giving the complexity of the simulated area, and an increase in the resolution would only amplify the bias coming from the parent domains. Only downscaling the model up to 250 m does not guarantee a better simulation or representation of the valley-ridge circulation. Thus, I believe the authors should try to quantify these uncertainties before use as an input for cloud parcel model and better explain some of the settings for the WRF simulation. Why do the authors use the MYJ PBL scheme to represent a convective regime over complex terrain? Is there a specific reason for using a local scheme rather a non-local scheme for a convective regime? Do the authors use higher resolution surface information in the 250m simulation to address the increased resolution in the model simulation? Minor points: Page 4 – line 12: "The model will be made avai";Page 18 – line 16: "wam"

---

## Referee Comment (RC4) · Anonymous Referee #4 · 16 Jul 2018

This study introduces a cloud parcel model (CPM) which includes aerosol activation, droplet condensation, collision-coalescence, and entrainment to study the cloud-aerosol interaction in cumulus congestus clouds. A case study that used the aircraft measurement and WRF simulations as the input parameter and initial conditions was studied. Sensitivity studies on the parameters were conducted. The conclusion was that the cloud development was sensitive to the aerosol concentration, entrainment, condensation coefficient but not sensitive to the hygroscopicity. Finally, limitations of the study were analyzed.

General remarks (Page = P, Line = L):

[Figure]

1) Overall, the manuscript is not well-organized. First, the objectives of this study are not clearly present to readers. Second, the descriptions and analysis of the experiments were spread out in the manuscript, appendix, and the supplementary information. For example, part of the measurement data was introduced in the manuscript, and some others were described in the supplementary information. But the authors referred both back and forth (esp. when referring to the figures), which made the readers hard to follow coherently. I suggest the authors either merge the same content to the manuscript or to the supplementary information. Same suggestions go to the appendix.

2) The content in the "summary of sensitivity analysis" and "summary and discussion" are redundant and most of the contents can be found in both sections.

3) The author used "as expected" many times throughout the article (e.g., P 11 L 11, P17 L8, P19 L25), while the physical explanation behind the "expected" results was very obscure. I suggest the author remove such statements or bring out explicitly the explanation.

4) The scientific contributions of this study are not clear. I feel the paper mainly focused on building up the model, testing the sensitivity of the parameters and confirming the test results through comparison with previous studies. It is suggested that more effort should be put into giving physical explanations.

Specific remarks:

1) The author mentioned E_coal in equation (11), but what value was taken and what assumption was made? It was not mentioned in the paper.

2) The author mentioned cloud "core" several times, but what defines the core? Was it defined by the updraft speed exceeding a certain value?

3) On P11 L11, the author states that "droplet spectra in stronger updrafts at the core" have "narrower size range compared to the samples at the edge of the cloud". This

seems to contradict to the founding in the P3 L13 in the supplementary information that stronger updrafts lead to a wider spectrum. Does this suggest that both entrainment and strong updraft help broaden the size spectrum? If so, the comparison between strong in-cloud updraft region and weak updraft in the cloud edge was not sufficient to give any physical implication.

4) In section 4.2.2, the author stated implicitly that the entrainment strength was inversely proportional to the cloud parcel size R, but the accurate formulation of entrainment strength was not shown.

5) In P18 L20, the author relates cloud base height to the scale height, Hs. How was Hs calculated?

6) The last paragraph in P18 was mainly a review of the previous studies, this should be moved to the introduction, otherwise, the authors are suggested to make connections with the results of the present study.

Based on the comments above, I suggest the paper is not suitable for publication on ACP unless all the above concerns have been addressed.

---

## Author Response (AR1)

**Dear Dr. Hailong Wang**

Please find attached item-by-item Replies to Reviewers. We believe we addressed the Reviewers concerns exhaustively. In particular we should like to highlight the following points:

a) The question of Journal Fit, that is appropriateness of the manuscript for ACP raised by one Reviewer -

We respectfully disagree. ACP has a long record publishing parcel model development (e.g. Antilla and Kerminen, 2007, Antilla et al., 2012, Kuba and Fujiyoshi, 2006, Kumar et al., 2009, Pringle et al., 2009, Reutter et al., 2009, Vali and Snider, 2015). Many of these are based on much simpler scenarios than what is proposed here and includes cases that only focus on parameterization development and studies that perform parametric sensitivity studies using parcel models without any observational constraints. Application of a parcel model in complex terrain, including entrainment and cloud microphysics in the model, comparing with WRF, and being constraint by aircraft and ground observations of aerosol and microphysical properties are clearly new science thrusts that go beyond standard parcel model papers. Based on this we argue that the work is clearly appropriate for ACP.

b) Paper organization and specifically the use of Supplemental Information, which required that Reviewers either print or have open an additional window to review the manuscript.

We respectfully disagree with the basis for concerns regarding organization based on the argument that not all material is provided in the main manuscript. Instead we followed well accepted practice in all high quality journals with primary information including model governing equations and description of field measurements in the main manuscript, whereas secondary equations are provided in Appendix. All support material necessary to provide readers with information necessary to check details and replicate the work is provided as Supplemental Information. Field observations are all publicly available from the NASA GPM GV website. We are making the model publicly available and welcome scrutiny, which requires that all relevant steps of the study be documented. Nevertheless, we recognize the manuscript presents a very complex study and we made a conscientious effort to reduce references to Supplemental material to a minimum.

c) Clear articulation of context for the work -

We added a paragraph to Introduction to explain the scientific context of this study and revise other paragraphs for clarity and to emphasize our long-term view that understanding aerosol-cloud processes is one critical step toward understanding aerosol-cloud-precipitation processes in mountainous regions, an in particular with regard to the time-scales and spatial heterogeneity of seeder-feeder interactions, and its implications for precipitation estimation from satellite remote-sensing and prediction using NWP.

d) Significance of the work - We make the following points:

1) Since when we first published this model in ACPD (Duan et al. 2017), we are not aware of any could parcel model (CPM) published by then that can simulate entrainment, condensation and collision-coalescence processes and produce vertical distributions of all processes and particle sizes. Relying on the open publication of our model in ACP Discussions to develop another model does not invalidate this point, and it is expected that this be recognized by appropriate use of citations. In addition the DCPM model will be shortly available from *github*. Further, if the manuscript is accepted for publication, the code can be published as supplement to the paper, and is already available upon request, and we hope to see it widely used and tested;

**2)** It is the first time in the peer-reviewed literature as far as we are aware that a CPM was used to simulate realistic conditions with concurrent multiple observations available for cross-check and analysis generally, and this is particularly challenging in complex terrain;

**3)** The sensitivity analysis indicates that widely used estimates of the condensation coefficient used to interpret field campaign data in tens if not hundreds of papers in the literature (some of which are cited in the manuscript) could be up to one order of magnitude off the correct value in realistic cloud environments, because the standard value used (ac=0.06) is the average value obtained from highly idealized laboratory chamber landmark experiments for a single suspended drop (Shaw and Lamb, 1999).

**4)** Because the CPM can simulate entrainment using different approaches, we can illustrate the highly nonlinear interplay of physical and chemical processes, and we make the very important point that it is not possible to probe aerosol-cloud interactions if the interactions among these processes are ignored, and

**5**) the simulations and evaluation against observations clearly show that the interplay of entrainment and growth processes, and in particular

collision-coalescence processes, play a key role in spectral broadening of cloud drop spectra with height potentially explaining bimodal distributions and the development of droplet sizes that can grow to become raindrops. It is the first time that the competitive interplay of entrainment and microphysics is clearly exposed without ambiguity. Further, it also explains the heterogeneity of samples at the same cloud height consistent with heterogeneous updrafts and thus time-scales of advection vis-à-vis microphysics. This is now articulated and emphasized in the Abstract for the final revised paper.

These and other minor comments were addressed. We note that no technical changes were made to the manuscript because none were warranted, but we made a conscientious effort to improve the reading according to Reviewer's suggestions.

We thank you for your consideration,

Ana P. Barros Markus Petters Yajuan Duan

Reply

We were not able to reply to the Reviews earlier for personal reasons, and we appreciate the Editor's patience extending the deadline. Below are item-by-item Replies to Reviewer #1 in blue to distinguish our replies from original Reviewer statements.

This study uses a warm-phase cloud parcel model that simulates the cloud droplet activation by aerosol particles, water condensation, collision-coalescence, and lateral entrainment processes to investigate aerosol-cloud interactions in one of the IPHEx cases. The comparisons between the in-situ observations and parcel model sensitivity results indicate that the condensation coefficient is the most important parameter determining cloud droplet number concentrations, liquid water content and size distribution. The cloud development is also sensitive to entrainment and aerosol concentration at cloud base but is not sensitive to aerosol hygroscopicity.

The manuscript is not very well organized. Readers have to resort to different locations

(main contexts, appendix and supplemental materials) for important and necessary information.

We appreciate the Reviewer's efforts to conduct a thorough review by checking all materials provided. We respectfully disagree regarding the paper organization. Use of Appendices and Supplementary materials is standard practice in many high-quality journals, and it is necessary to protect the expectation that results can be replicated by anyone who wants do so. This includes providing partial results and full details of model configuration, initial conditions and observations used to examine model results and interpret the integrated model and observations. The alternatives at hand are (i) to bring everything in to the main paper, which then becomes a project report, (ii) to suppress information, or finally (iii) to include it in a way that readers interested in specific details may be able to check those out. Another alternative is to divide a paper in several parts and publish several papers, which does not reduce the inconvenience of having to resort to different locations for information, even if it does artificially increase the authors' productivity.

It is indeed a struggle to present all information that is relevant to a comprehensive study like the one presented here. Even more so in this day and age when there is substantial pressure to publish short and to-the-point papers. The manuscript was revised before as per the Author's Comment uploaded when the present manuscript was published in ACP Discussions, and we addressed organization issues related to Appendices and Supplemental materials while maintaining the goal of providing all information relevant to replicate the study. Again, we stress the point that is a fundamentally good scientific practice to provide all information necessary for others to replicate the work and check assumptions. We are confident that this is the case here.

The development of a cloud parcel model is a non-trivial work. However, such a tool should be used to answer critical scientific questions. There are few new scientific points being discovered in this work, which does not makes it qualified for the ACP publication. I would suggest the authors to submit this manuscript to other journals that have an emphasis on model development or test.

We appreciate the Reviewer's recognition that developing a new cloud parcel model is not trivial, and we respectfully disagree with the Reviewer's assessment regarding the paper's value. As the Reviewer recognizes the paper does make new contributions even if, as stated by the Reviewer, some are related to model development. It is not possible to probe critical scientific questions in many cases without models, and in particular without models that do not capture fundamental physics and chemistry. Surely, it is not more scientific to use poorly documented and, or incomplete and weak models to address critical scientific questions, and then publish the results ignoring the underlying model's handicaps. Specifically, regarding this manuscript, we make the following points: 1) since when we first published this model in ACPD (Duan et al. 2017), we are not aware of any could parcel model (CPM) published by then that can simulate entrainment, condensation and collision-coalescence processes and produce vertical distributions of all processes and particle sizes. Relying on the open publication of our model in ACP Discussions to develop another model does not invalidate this point, and it is expected that this be recognized by appropriate use of citations. In addition the DCPM model will be shortly available from *github*. Further, if the manuscript is accepted for publication, the code can be published as supplement to the paper, and is already available upon request, and we hope to see it widely used and tested; 2) it is the first time in the peer-reviewed literature as far as we are aware that a CPM was used to simulate realistic conditions with concurrent multiple observations available for cross-check and analysis generally, and this is particularly challenging in complex terrain; 3) the sensitivity analysis indicates that widely used estimates of the condensation coefficient used to interpret field campaign data in tens if not hundreds of papers in the literature (some of which are cited in the manuscript) could be up to one orders of magnitude off the correct value in realistic cloud environments, because the standard value used (ac=0.06) is the average value obtained from highly idealized laboratory chamber landmark experiments for a single suspended drop (Shaw and Lamb, 1999). 4) because the CPM can simulate entrainment using different approaches, we can illustrate the highly nonlinear interplay of physical and chemical processes, and we make the very important point that it is not possible to probe aerosol-cloud-precipitation interactions if the interactions among these processes are ignored, and **5**) the simulations and evaluation against observations clearly show that collision-coalescence processes play a key role in spectral broadening of cloud drop spectra with height potentially explaining bimodal distributions and the development of droplet sizes that can grow to become raindrops. This is now articulated and emphasized in the Abstract for the final revised paper.

I listed some of my concerns in the following: 1. If the condensation coefficient is the dominant parameter for cloud development and evolution in the early stage, how do the authors choose a value or develop a parameterization to provide a reasonable value for the more detailed 3D simulations?

An important contribution of this manuscript is to clearly make the point that these fundamental coefficients are not settled, and therefore further research is required. Our advice to anyone interested in 3D modeling of clouds is to conduct a careful evaluation of their model against direct and indirect observations and to conduct a multivariable uncertainty analysis to guide parameter specification, and indeed we would propose that the accommodation parameters be adaptive and change depending on aerosol type and environmental conditions. This is an entire new research project, and solving the science problem we identify and document in this manuscript is out of the scope of any possible revisions.

2. As the authors pointed out, lateral entrainment is not appropriate especially for orography influenced clouds. A more appropriate en- trainment scheme might be needed for the work.

The interpretation is not accurate. What we are saying is that in complex terrain, the 3D nature of circulations make for complex transport features that cannot be represented by the simple lateral entrainment. Further, lateral entrainment is simplistic in unstable environments with complex cloud morphology, as the conceptual model is that of a perfectly 1D, symmetric, buoyant plume.

Indeed, developing better entrainment models is a great need in the wide scientific community, but it out of the scope of this paper. By including entrainment in our CPM and by testing different parameterizations, we document and expose an important scientific challenge. There is an entire community of turbulence experts working on developing other parameterizations. Schemes currently used in most NWP and Atmospheric Chemistry Models are either much simpler or of the same level of complexity. We heed the Reviewer's advice and highlight this need in the conclusions.

Equation 8 is the transport equation applied to each bin (particles of specific size), and it is necessary to preserve the conservative property that is the net mass balance in each bin while accounting for the fact that different particles sizes (different mass and velocity) will be entrained at different rates.

4. Page 10, lines 14-15, why the corrected CDP spectra that is shifted to smaller size provided confidence in the performance of the CDP probe during the IPHEx campaign?

Because it corrects for the overestimation errors of measuring several particles at the same time as explained in the beginning of the paragraph. (Page 10, Section 3.2).

5. Page 11, line 31, the volume ratio of 1.026 is not correct. It should be 1.021.

This value is correct as implemented in the model. Note that the model grid is dynamic and changes as needed to accommodate new droplet sizes as they form. We should have stated up to 1000 bins. S suggested by another Reviewer we added Tables3a and 3b to clarify model configuration and parameters used.

6. Figure 7b looks wrong to me. The condensation process is inversely related to the droplet size. But in Fig. 7b, the entire DSD shifts to the right without showing any narrowing of the DSD.

Figure 7b is correct. It shows the DSD evolution with height reflecting the nonlinear interactions among entrainment and droplet growth processes incluing condensational growth and collision-coalescence processes. As described in Section 2, entrainment of dry air in this model does not only affect temperature and liquid water content, and thus updraft velocity and saturation, but it also affects the spectra, and specifically dry CN (Eq. 8). Therefore, the spectra does not narrow necessarily as new in-cloud CN depleted by activation are replenished by entrainment and previously activated drops continue to grow, and as they become larger the efficiency of collision and coalescence processes increases significantly thus broadening the spectra potentially lead to the development of a coarser mode such as found in the observations, and pointed out by Reviewer #2 also.

7. Since many parameters impact the cloud development, there are multiple combinations of these parameters to provide the same cloud development trajectory. How can the authors justify which combination is the right one?

The Reviewer raises a point that is pertinent when there are no observations to guide and support interpretation of model results. However, this not the case here. Indeed, we present in the manuscript results from over 30 different combinations of parameters (new Table 3b), and there is clear convergence to specific values and ranges of parameters compared against IPHEx observations. This was addressed explicitly in the Conclusions (last paragraph). In particular, the next step is to map the joint nonlinear dynamics of entrainment and condensation processes, which we are already working on. This implies identifying the physically-meaningful regions/regimes of parameter-space that could be used to guide parameter specification in 3D models. Indeed, this and in itself justifies the importance (and need) for a model such as the DCPM that enables pursuing these research questions. We revised the writing in the last paragraph in Section 5 to improve clarity as shown below.

"In the present study, the local sensitivity of selected model parameters are assessed individually over certain ranges based on IPHEx data and the literature. Overall, thirty different sensitivity simulations corresponding to different parameter combinations were conducted, and the results suggest that the ranges of parameters that lead to physically-meaningful results consistent with observations are small. Nevertheless, the present study underlines the importance of the relationhsip between entrainment processes that determine the local- (microscale) and cloud-scale thermodynamic environment around individual particles, and the aerosol condensation coefficient that measures the effectiveness of condensation processes in the same thermodynamic environment. Given the multiscale thermodynamic structure of clouds, these interactions suggest that realistically the condensation coefficients in the natural environment are transient and spatially variable. Therefore, further research is necessary to map the joint nonlinear dynamics of aerosol-cloud interaction parameters to identify physically meaningful regimes in parameter-space, and to link these regimes to environmental conditions and aerosol properties toward reducing related uncertainties in the estimation of the aerosol indirect effect. Ongoing work is focusing on exploring the sensitivity of the DCPM in a multi-dimentional parameter space to quantify multiple parameter interactions (Gebremichael and Barros, 2006; Yildiz and Barros, 2007) on aerosol-cloud processes using the fractorial design method (Box et al., 1978)."

Reply

We were not able to reply to the Reviews earlier for personal reasons, and we appreciate the Editor's patience extending the deadline. Below are item-by-item Replies to Reviewer #2 in blue to distinguish our replies from original Reviewer statements. We thank the Reviewer for very carefully checking the manuscript and offering very helpful comments and suggestions.

In this study, a new advanced cloud parcel model, which includes description of condensation, collision-coalescence, and lateral entrainment processes, is developed. The model is designed so as to be coupled with the rainfall microphysics column model describing the evolution of raindrops size distributions. In wide aspect the model will be used for investigation of aerosol-cloud interaction. In order to set initial and boundary meteorological and aerosol conditions for model study authors also analyzed the results of measurements, obtained in the IPHEx campaign in this paper. The results of the supplemental simulations by the WRF model are also applied to specify sounding, used in the parcel model. The data obtained in the IPHEx campaign also serves as a reference in the comparison of model and measured results.

The main output of the study illustrated in the figures 6-10 and B1-B3, is the results of sensitivity simulations, that show the variability of the different microphysical and thermodynamic quantities such as droplet concentration profiles, LWC profiles, supersaturation profiles, droplet size distribution (DSD) to variation of condensation coefficient, to type of parcel model (bubble vs jet), to initial aerosol concentration and hygroscopicity parameter as well as to environment conditions and initial updraft velocity. Comparison with experimental data allowed to choose optimal parameters providing minimal discrepancy between model and experimental data. Agreement between modeled and measured DSDs can be characterized as reasonably good.

**Thank you for the comprehensive and positive assessment.**

Several corrections which fall between "major" and "minor" should be done before the manuscript is published in ACP:

1. I think the purpose of this study must be specified more clearly. If this study is a part of a more general study an introduction to the general purpose should be done.

Thank you for the suggestion. We revised and expanded the following paragraph in Introduction (bottom of Page 2, and top of Page 3 in revised manuscript):

"In the Southern Appalachian Mountains (SAM, Fig. 1), persistent low-level clouds and fog (LLCF) play a governing role in warn season rainfall by increasing the frequency and duration of light rainfall and drizzle, and by enhancing storm rainfall

via seeder-feeder interactions (SFI; Wilson and Barros, 2014, 2015 and 2017; Duan and Barros, 2017). Albeit with large spatial variability, microphysical observations and idealized model simulations of the dynamical evolution of raindrop size distribution (RDSD) with height show that SFI in the lower atmosphere can explain up to one order magnitude increases in rainfall rate at low elevations similar to orographic enhancement at higher elevations in the SAM. Understanding and modelling the spatial variability of the vertical microstructure of clouds in complex terrain is therefore key to understand precipitation processes toward improving rainfall estimation and prediction. Whereas previous studies linked LLCF in the SAM to high biogenic aerosol loading produced locally with occasional influx from remote pollution sources (Link et al., 2015; Lowenthal et al., 2009), quantitative understanding of the indirect effect of aerosols on clouds with implications for precipitation dynamics including SFI is lacking. The purpose of this study is to investigate ACI in the SAM integrating models and observations collected during IPHEx (Integrated Precipitation and Hydrology Experiment; Barros et al., 2014) with a focus on the evolution of cloud droplet spectra with height. This is an important first step toward understanding the spatial variability in the vertical structure of cloud microphysics from one location to another toward capturing the observed spatial and temporal heterogeneity of SFI."

In addition, we reorganized the text to discuss two different modeling approach alternatives NWP-type and CPMs. Then we explain the overall goal of simulating end-to-end ACPI and impact on radar reflectivity and quantitative precipitation estimation (QPE).

Three microphysical processes (condensation, collision-coalescence, and lateral entrainment) are described very accurately in the model. At the same time such processes as droplet sedimentation and ventilation are completely absent from the model. Authors should justify the omission of these processes.

The differential sedimentation is very important and it can be accurately simulated using an Eulerian-Lagrangian formulation (see Pratt and Barros, 2007; and subsequent papers). The reason why it is not explicitly addressed here is because in the context of cloud development, the vertical velocity of updrafts (V> 1m/s) always exceeds the fall speed of the droplets (V

Reply

We were not able to reply to the Reviews earlier for personal reasons, and we appreciate the Editor's patience extending the deadline. Below are item-by-item Replies to Reviewer #3 in blue to distinguish our replies from original Reviewer statements.

This study presents a new entraining cloud parcel model that includes activation, condensation, collision-coalescence, and lateral entrainment processes. This model was applied to a case study to investigate dominant factors in determining the microphysical development of clouds over complex terrain such as aerosol-cloud interaction during IPHEx campaign. The model was tested for a mid-day cumulus congestus case where aircraft measurements were available. Also, the authors used the measurements from IPHEx campaign and WRF modeling to provide initial conditions of some variables to the cloud parcel model. The authors stated that the modeling results for the reference simulation achieved a good agreement with the cloud droplet number concentration, liquid water content, and droplet size spectra observation few meters above cloud base.

Based on in-situ measurement and model sensitivity results, the authors found that condensation coefficients are a key parameter for this case. In addition, the model is sensitive to entrainment and aerosol concentration at the cloud base. Although the authors performed a good job in the development and analysis of the new model, the manuscript did not present a clear focus on answering new scientific questions related to the aerosol-cloud interactions and in some parts of the manuscript the goal of this work can be misinterpreted as a model development, which makes the current status of this manuscript not scientific significant for ACP publication.

We respectfully disagree with the referee about the appropriate scope of the work for ACP. ACP has a long record publishing parcel model development (e.g. Antilla and Kerminen, 2007, Antilla et al., 2012, Kuba and Fujiyoshi, 2006, Kumar et al., 2009, Pringle et al., 2009, Reutter et al., 2009, Vali and Snider, 2015). Many of these are based on much simpler scenarios than what is proposed here and includes cases that only focus on parameterization development and studies that perform parameteric sensitivity studies using parcel models without any observational constraints. Application of a parcel model in complex terrain, including entrainment and cloud microphysics in the model, comparing with WRF, and being constraint by aircraft and ground observations of aerosol and microphysical properties are clearly new science thrusts that go beyond standard parcel model papers. Based on this we argue that the work is clearly appropriate for ACP.

Further, as the Reviewer notes, the paper is not just about the model development, but the model is the platform that enables a detailed process study constrained by observations that challenges common assumptions in the interpretation and extrapolation of field observations using standard CPMs. The results from sensitivity analysis with regard to the condensational growth indicate that the best agreement between simulated and observed spectra at flight level are lower than the typical value used for aerosol-cloud droplet closure CPM studies at cloud base and without entrainment. This is consistent with the range of values in the peer-review literature for a wide range of experiments for realistic laboratory set-ups (see literature cited). By enabling targeted sensitivity analysis with regard to entrainment and demonstrating the competitive nonlinear feedbacks between condensational growth and entrainment on the vertical structure of cloud droplets which in turn affects cloud development, this study brings up the need to improve physical parameterizations of entrainment and the need to reanalyze data from field experiments that neglected these interactions.

Finally, we respectfully disagree with the Reviewer that presenting all the details in the formulation of a new model and demonstrating that it works well and indeed meets very high standards of rigor against other existing models would not be in itself a significant contribution vis-à-vis addressing science questions with incompletely documented or unevaluated models, and without providing enough any details to enable an independent researcher to reproduce the findings reported. Rather, thorough and transparent documentation and reproducibility are essential to the scientific enterprise.

As such, I would recommend major revisions.

Specific comments: 1) The manuscript is not well organized, in some parts of the manuscript the readers have to go back and forth between main text and supplementary material, which make the readers confused on whether the information is important or just a supplementary information, one example is the section 3.3 lines 24-31. I would suggest the authors find a more organized way to present those discussions and a better transition between the text, the main figures and the supplementary figures.

The Reviewer's point regarding Section 3.3 is well taken. The Section was revised and substantially shortened, and references to supplementary information were reduced to two locations only.

2) Page 15 lines 18-19 the authors stated that one of the explanations for the absence of small drops is the uncertainties coming from WRF simulated radiosondes. I can see a possibility that the WRF simulation did not represent correctly the vertical structure of the atmosphere (especially in the lower levels) giving the complexity of the simulated area, and an increase in the resolution would only amplify the bias

coming from the parent domains. Only downscaling the model up to 250 m does not guarantee a better simulation or representation of the valley-ridge circulation. Thus, I believe the authors should try to quantify these uncertainties before use as an input for cloud parcel model and better explain some of the settings for the WRF simulation.

The settings for the WRF simulation are described in detail in Section S3. In order to account for these uncertainties, the ensemble mean of 6 WRF profiles in the valley around location IC was used. This writing in the main text was revised for clarity (Lines 5-10, page 12). Further, the "optimality" of these settings for the region of study are well documented in the peer –review literature (Wilson and Barros, 2015, 2017; Sun and Barros, 2013 and 2012).

Why do the authors use the MYJ PBL scheme to represent a convective regime over complex terrain? Is there a specific reason for using a local scheme rather a non-local scheme for a con- vective regime? Do the authors use higher resolution surface information in the 250m simulation to address the increased resolution in the model simulation?

The configuration used for WRF in these simulations is the result from extensive and detailed studies reported in Wilson and Barros (2015 and 2017) and Sun and Barros (2012 and 2013). It turns out that the MYJ PBL scheme is appropriate here because it is local and is more sensitive to landform heterogeneity that is critical in complex terrain. Indeed, this is also the case for explicit convection simulations when capturing localized land-atmosphere interactions is important (e.g. Erlingis and Barros, 2014). Yes, higher resolution terrain data are used at 250 m obtained by fractal downscaling to preserve spatial structure following Bindlish and Barros (1998). See also Wilson and Barros (2015). In addition to higher terrain resolution, by bringing the model resolution down to 250 m, there is a significant increase in model effective resolution, that is the velocity scales resolved by the model (e.g. Nogueira and Barros 2014; Sckamarock, 2008).

Minor points: Page 4 - line 12: "The model will be made avai";Page 18 - line 16: "wam"

The grammar and spelling were fixed. Thank you.

The Reviewer's point is well taken. The paragraph was shortened and revised to focus on the results of the present study to emphasize the theoretical and physical basis of the results. Section 4.2.5, Page 18 and 19.

Based on the comments above, I suggest the paper is not suitable for publication on ACP unless all the above concerns have been addressed.

All comments raised were addressed. Thank you.

С3

[revised manuscript text omitted]

---

## Author Response (AR2)

Dear Dr. Hailong Wang

Please find attached item-by-item Replies to Reviewer.

The Supplemental information, Figures and Appendices did no change. The main manuscript was edited thoroughly to address Reviewer concerns with writing. We submit both marked and unmarked revised versions.

Thank you for your consideration,

     Ana P. Barros
     Markus Petters
     Yajuan Duan

**Replies to Reviews**

Review comments are in black.  The Replies are in Blue.

The revision addressed most of the questions raised by the reviewers and the overall organisation of the manuscript is improved. However, there are still some questions remained ambiguous. Besides, the response letter is not well-written. The sentences are very long and redundant and sometimes bear severe grammar mistakes that one cannot follow. Therefore, I do not support the publication of the present form of the manuscript before it is carefully revised based on all comments made by the previous and current reviews.

We appreciate the Reviewer's careful reading of the manuscript.  We regret any grammar errors in the response letter.  We agree that clarity is very important, and we make a conscientious effort to address the issues raised as detailed below.

(1) The revision is not very well-written. Some sentences are very long and difficult to understand. For example:

We shortened the sentences and made an effort to improve clarity throughout as the Reviewer can see from the marked manuscript. Matters of writing style are personal however, not technical, and indeed we expect that this remains apparent to some degree in the manuscript reflecting its authorship.

a. The last sentence of the abstract: "Nonetheless, the model simulations provide new constraints of the determinant factors of convective cloud formation leading to mid-day warm season rainfall in complex terrain." I think the sentence is either incomplete or needs structural improvement.

The sentence was dropped in the revised Abstract.  A statement was added regarding the need to incorporate the impact of entrainment on the vertical evolution of the cloud droplet spectra.

b. Line 15-17, page 3: " The representation of physical and chemical processes related to clouds and precipitation in numerical models relies on parameterizations with varying degrees of uncertainties depending on space-time model resolution because of multiscale and complex physics, (Khairoutdinov et al., 2005; Randall et al., 2003)" The sentence is wordy and not understandable, and the comma should be period.

The sentence was revised to read:

"Clouds and precipitation are represented in numerical models using parameterizations of physical and chemical processes with varying uncertainty that depends on model temporal and spatial resolution (Khairoutdinov et al., 2005; Randall et al., 2003)."

c. Line 32, page 19 – line 4, page 20
I would suggest splitting all sentences that exceed 3 lines, if possible.

The sentence was split into shorter sentences. The entire manuscript was edited in the same spirit.

(2) One critical comment that shared by all reviewers is that the paper did not present a clear scientific contribution. The revision still did not satisfactorily provide convincing arguments. Specifically, the scientific purpose and the deliverables of the study is very vague. What is the research question of this study? What is new? What is confirmed from this study and what is the conclusion? What is the contribution of this study? The readers should be able to obtain with ease the answers to the above questions from the paper. In the response letter, the authors answer three times regarding the significance of the work. However, the statements are either too general (i.e., point (4)) or too wordy to understand (i.e., point (3)) or both (i.e. point (5)). I can marginally get from the first two points that the model provides a new tool to study ACI that considers complexed terrain, constraints from the observations, and entrainment. I would suggest the authors use more clear statement to answer the above questions.

The abstract and manuscript were revised for clarity.  We respectfully disagree that the statements are too general.  Point 4 highlights the fact that lateral entrainment is accounted for in the new CPM.  Point 5 states that interactions among lateral entrainment, condensational growth and collision-coalescence processes are essential to capture spectral changes with height observed during IPHEx flights well above cloud base in complex terrain.

The new model to study ACI simulates interactions among lateral entrainment, condensational growth and collision-coalescence processes as parcels rise.  The numerical experiments reveal that these interactions are key to capture the amplitude and broadening of cloud spectra with height in the cloud.  Therefore, these processes are important to include in parameterizations of convective clouds to simulate realistic vertical structure during the early stages of cloud development.

This study adds new insights to the previous body of the literature analyzing field campaign data using adiabatic CPM models (thus neglecting lateral entrainment) without collision-coalescence applied to simulate number concentrations near cloud base.

(3) The authors stressed more than once in the manuscript as well as in the response letter that their model can illustrate the interplay of physical and chemical processes. However,

based on the model description, there is no chemical processes involved in the model. The only thing that is related to "chemistry" is that the model considers the chemical component which affects the hygroscopicity of the aerosol particles. This in no circumstances can be defined as a chemical process. It is strongly suggested that the authors remove the misleading statements that appear in the manuscript.

The referee is correct that there are no chemical processes in the model. Unfortunately, it is unclear what misleading sentences in the manuscript the referee is referring to. We searched the text for "chemical", "composition", "physicochemical" and "chemistry". In all places it is clear that the model response is to hygroscopicity, which is determined by chemical composition. The sentence stating "interplay between physical and chemical processes" was poor wording and only appeared in the response to the referees. A more precise statement would have been "interplay between aerosol number, size and hygroscopicity on cloud properties."

(4) The idea of "seeder-feeder interactions" is used quite frequently without any explanation of what it is. I suggest the author give a brief and clear description on what is the seeder and what is the feeder and how the mechanism works. Simply mentioning the name does not help to clarify the role of LLCF in rain enhancement.

A short definition was added in the Introduction section following the first use of the terminology.

"SFI refers to the modification of cloud and rain drop size distributions when precipitation from above (seeder clouds) falls through lower cloud layers (feeder clouds) to significantly enhance drop collision-coalescence efficiency, and consequently rainfall rates."

(5) What is ACPI? It not defined in the manuscript.

It is now defined.  It refers to Aerosol-Cloud-Precipitation Interactions.

(6) Further questions in addition to the 3rd question by Reviewer#1: In equation 8, the droplet in each bin is associated to the aerosols of its corresponding bin. However, why is the droplet number concentration only determined by the number concentration of the aerosol of the ith bin? It is quite possible that aerosol of other bins can contribute to the droplet of ith bin. I.e., droplet of same size (wet radius) can have very different dry radius.

Equation 8 describes changes in number concentration in each bin due to entrainment alone. At a given time-step, the total changes in number concentration in all bins include changes not only due to entrainment, but also condensational growth and collision-coalescence.  For clarity, the equations that describe each process are laid out in different subsections of Sect. 2. Indeed, with time, the drops move from one bin to another as the wet radius changes.  This is clearly stated in the introduction to Section 2:

"…a moving grid structure is implemented so that an initial size distribution based on a fixed grid discretization can change with time according to the condensational growth. This approach allows particles in each bin to grow by condensation to their exact transient sizes without partitioning between adjacent size bins. Subsequently, collision and coalescence are resolved on the moving bins that evolve from condensation."

(7) In the answer to the 1st question of reviewer#2, the authors stated that the fall speed of the droplet is < 0.1 m/s. However, this statement is not true. If the model includes drops up to 7mm (line 20, page 7), the fall speed shall easily exceed 1 m/s (Fig. 6 in Beard 1975). Therefore, the statement that the sedimentation effect is not important because "the updraft speed always exceeds fall speed" is not valid. In addition, since the parcel is rising adiabatically (i.e., the model is in Lagrangian framework), the droplet sedimentation should not be affected by the updraft.

We believe the Reviewer refers to Beard (1976) also cited in the manuscript. We have now added three other references to the manuscript: Beard (1977) that shows relevant figures and tables for easy verification, and Guzel and Barros (2001) and Barros et al. (2008) that report controlled tower experiments to measure drop terminal velocity.

The goal of the coupled parcel-rainshaft modelling framework described in Section 1 is to simulate microphysics from the time of CN activation to the time raindrops reach the ground. However, in this manuscript, the parcel model is used strictly to simulate the early stages of cloud development and therefore simulated (and observed) droplet diameters remain below 40 μm, and thus terminal velocities do not exceed 0.06 m/s (e.g. Beard, 1977). This is one order of magnitude smaller than updraft velocity. In addition, for the particle sizes in question the time scales of condensation (milliseconds, see model time-step) are much shorter than the time-scales to reach terminal velocity, which vary inversely with particle size. Small drops oscillate and "wobble" for a long time before they reach terminal velocity. This means that, as drops are getting wetter, and thus growing, they are in a transient state and may never reach terminal velocity.

In any case, as stated in the revised manuscript, Eulerian-Lagrangian simulations using the terminal velocities for the respective bin sizes increases simulation times by several orders of magnitude without producing different results, and therefore Lagrangian sedimentation processes were by-passed in this specific context.

All our simulations including rain-sized drops explicitly simulate differential sedimentation.

The sedimentation sentence in the Introduction to Section 2 was removed. Instead, a more comprehensive statement was added to Section 2.3 on numerical formulation and implementation (Lines 22-30, Page 8).

(8) To the 3rd question by reviewer #4: no change was made in the revised manuscript.

A search for statements using "as expected" and similar terminology indicates they were all removed from the revised manuscript.

(9) Spelling & grammar:
a. Line 32, page 2: warm-season
b. Line 26, page3: is adequate to…
c. Line 8, page 12: was used in the process-study
d. Line 1, page 20: bin microphysics
e. Line 31, page 20: parcel models
f. Line 3, page 21: aerosol-cloud interaction
g. Line 4, page 22: relationship
h. There should be space between references. i.e., Koren et al., 2008; Ramanathan et al., 2001.

All revised.

Thank you.

[revised manuscript text omitted]